# Mitigating Confounding Effects in Causal Discovery via Time-lagged Backdoor Pathways

## Abstract

Mitigating confounding effects is one of the fundamental challenges in causal discovery. This difficulty is amplified in more complex causal structures: where interactions involve colliders, mediators, and their hybrids, methods tailored for handling confounders may incur substantial errors, especially in the presence of latent factors. In this paper, we propose a novel causality discovery algorithm of Conditional Independence Test on Time-Lagged Backdoor Pathways (**CIT-TBP**). This approach intelligently leverages backdoor pathways induced by time-lagged causation to indirectly infer causal relationships, effectively eliminating the influence of various forms of complex interactions. Furthermore, by incorporating causal information flow, our method significantly reduces the impact of latent variables. We theoretically prove the rationality and effectiveness of the algorithm and experimentally validate it on several synthetic and real datasets. The experiment results demonstrate the superiority of our CIT-TBP against state-of-the-art methods. Compared with contemporary optimization-based methods, our causal discovery framework does not involve any black-box optimization process, and thus the causality derived are more direct and have a wide range of potential applications. The code is available at `https://anonymous.4open.science/r/CIT-TBP-F0E8`.

## 1 Introduction

Causal discovery (Bareinboim & Pearl, 2016; Arif & MacNeil, 2023) aims to uncover the underlying causal relationships from observational data, providing a foundation for reliable decision-making across diverse fields such as healthcare (Shi et al., 2019), economics (Pearl, 2009a), and social sciences (Sugihara et al., 2012; Arif & MacNeil, 2023). A central difficulty in causal discovery is addressing confounding variables that jointly affect treatment and outcome. The difficulty becomes even more pronounced when causal relationships exhibit time delays (Ye et al., 2015). For instance, in clinical medicine, there are temporal lags between medication administration and pathological indicator changes. Focusing solely on contemporaneous relationships between drug levels and pathological metrics may lead to erroneous conclusions. Therefore, mitigating confounding effects in the presence of time-delayed causal relationships remains a critical challenge.

Most existing methods are able to mitigate confounding effects in relatively simple time-series settings. However, their effectiveness deteriorates in more complex causal structures, where interactions involve colliders, mediators, and their hybrids (see Figure (1b) and Figure (1c)), and the presence of latent factors further exacerbates the problem. Constraint-based methods, such as PC-based algorithms, their temporal variant PCMCI (Gerhardus & Runge, 2020), and the latent-factor variant FCI (Entner & Hoyer, 2010; Malinsky & Spirtes, 2018), typically rely on the standard backdoor criterion to uncover causal structures. However, they often face challenges related to Markov equivalence classes, which make it hard to distinguish between confounders and mediators; that is to say, they prevent unique identification of complete directed acyclic graphs (DAGs). Time-series-based algorithms, such as Granger Causality (GC) (Barnett & Seth, 2014) and Rhino (Gong et al., 2023), often rely on assumptions about no latent factors and struggle to account for mediators, making it difficult to distinguish direct causal effects from indirect ones. Optimization-based approaches (Rolland et al., 2022; Bello et al., 2022; Gong et al., 2023) constrain the construction of causal graphs based on fundamental assumptions and inference rules. These methods typically target specific structural patterns guided by deep domain knowledge, rather than providing a comprehensive

treatment of causal architectures (Pearl, 2009a; Kampa & Castanas, 2008; Hartford et al., 2017). Moreover, temporal delays in causation frequently lead to erroneous inferences in such approaches (Castro et al., 2023; Biswas et al., 2023).

To address these challenges, we propose a novel two-stage causal discovery method (**CIT-TBP**) featuring an innovative time-lagged causal backdoor pathway concept. In the first stage, unlike traditional Backdoor Criterion (Pearl, 2009a; Xu et al., 2024) limited to contemporaneous data and vulnerable to V-structures (Geng et al., 2005), our temporal approach fully exploits backdoor pathways induced by time-lagged causal relationships; By effectively blocking time-lagged backdoor pathways that are unrelated to causality, we successfully achieve causal discovery across diverse causal interactions. Then in the second stage, we introduce the principle of causal information flow, which determines the existence of potential factors by the difference in entropy value. Theoretical proofs demonstrate our algorithm's effectiveness in eliminating common causes, chain structures, and colliders, even in the presence of latent factors. Without black-box pruning or optimization steps, it achieves more transparent and extensible causal relationships. Our contributions are three-fold:

a.) We develop a general causal discovery framework based on conditional independence tests, which introduces time-lagged backdoor pathways to eliminate the influence of confounding effects in different causal structures, including confounders, mediators, colliders, and more complex hybrids-thereby significantly improving the accuracy of causal structure learning.

b.) Combining statistical associations with causal information flow principles, our approach can infer causal graphs under latent confounders. Distinct from most of the optimization-based methods, our algorithm maintains full transparency without black-box operations.

c.) We evaluate the effectiveness of the algorithm through simulated linear and non-linear datasets and real-world data. We also validate the algorithm's effectiveness in mitigating the influence of latent factors. Results demonstrate the superiority of our CIT-TBP over baselines in both linear and nonlinear scenarios.

## 1.1 RELATED WORK

A fundamental way to discover causality is through interventions (Tigas et al., 2022; Zhang et al., 2023; Liu et al., 2023; Pearl, 2009a; Eichler, 2013). However, causality with interventions is costly and intrusive, which is often impractical, especially in cases with time lags. Topology-based sorting approaches (Wu et al., 2024; Rolland et al., 2022; Sanchez et al., 2023) utilize data log-likelihood for iterative node identification, but they may produce non-unique causal orderings and remain vulnerable to confounders. Score-based methods (Tsamardinos et al., 2006; Zhu & Chen, 2019), which optimize causal structures through acyclicity constraints and data fitting, also struggle with confounding factors, lack model interpretability, and perform poorly in the presence of time lags. GC-based methods (Sugihara et al., 2012; Gong et al., 2023), commonly used for time-series data , are highly susceptible to latent factors and struggle to fully eliminate the influence of confounding variables. Other time-series causal discovery algorithms, such as Cuts+ (Cheng et al., 2024a), are sensitive to noise and also rely on the assumption of no latent factors. Although methods (Entner & Hoyer, 2010; Malinsky & Spirtes, 2018; Jin et al., 2024) have been developed to mitigate the effects of latent factors, they often rely on strong assumptions about the underlying causal structures and can yield graphs with undirected edges. Constraint-based methods (Xu et al., 2024; Runge et al., 2019; Gerhardus & Runge, 2020) are hindered by Markov equivalence classes, which, in the presence of different causal interactions, lead to numerous undirected edges and causal equivalence classes during causal discovery, preventing the unique identification of DAGs. Inspired by these works, in this paper, we study causal structures in data with time-lagged causality and combine the principles of conditional independence tests and information entropy for eliminating the latent factors.

## 2 PROBLEM SETUP

## 2.1 MODEL DEFINITION

**Causal Discovery Model.** We focus on a Structural Causal Model (SCM) which consists of two components: causal graph and structural equation. A causal directed acyclic graph $\mathcal{G}$ consists of multiple nodes and directed edges, where each node in $\mathcal{G}$ represents a variable and the direction of

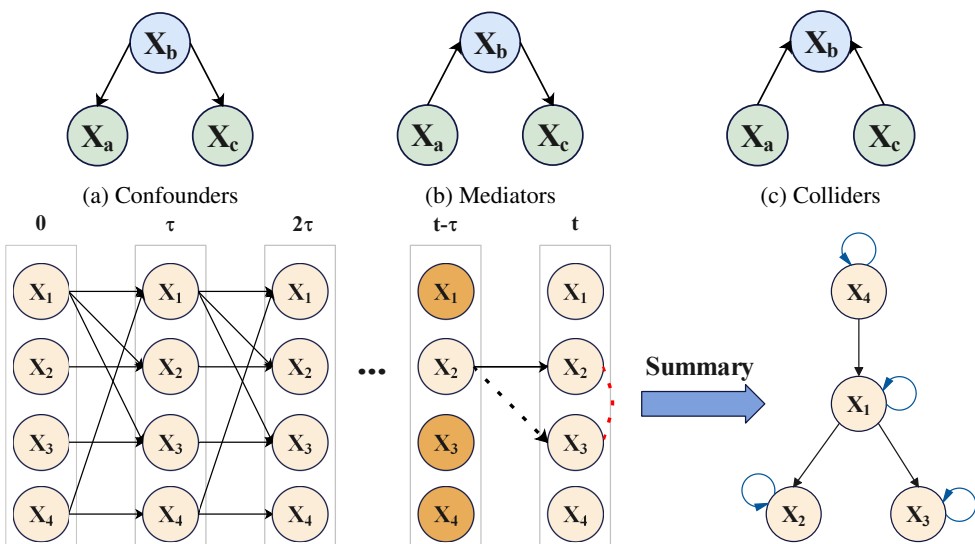

(a) Confounders      (b) Mediators      (c) Colliders

(d) Time-Lagged Backdoor Principle for eliminating confounders

Figure 1: (a,b,c) Three basic building blocks of variable interactions. (d) An example of the Time-Lagged Backdoor Principle on confounding factors: When we try to discover whether $X_2$ is an ancestor of $X_3$, $X_1$ acts as a confounding factor. There will be a spurious association between $X_2^t$ and $X_3^t$ (marked with a red dotted line) because of the non-causal backdoor pathway $X_2^t \leftarrow X_1^{t-\tau} \rightarrow X_3^t$. If $X_2 \rightarrow X_3$, $X_2^t \leftarrow X_2^{t-\tau} \rightarrow X_3^t$ will be a causal backdoor pathway. Therefore, when we get $X_2^t \perp\!\!\!\perp X_3^t \mid \{X_1^{t-\tau}, X_3^{t-\tau}, X_4^{t-\tau}\}$, we can conclude $X_2 \not\rightarrow X_3$. The right side displays a summary causal graph that abstracts away from the time-slice-specific causal graphs shown on the left.

the edges represents the causal direction. An n-variate time series can be denoted as $X = \{X_i^\tau\}_{n \times t}$, where $i \in \{1, 2, \cdots, n\}$ and $\tau \in \{1, 2, \cdots, t\}$. For example, $X_k^{t_a}$ represents the variable $k$ at time $t_a$.

**Definition 1** (Summary causal graph). *The summary causal graph (SCG) is a directed graph with an arrow from $X_i$ to $X_j$ ($i \neq j$) whenever there is a directed arrow from $X_i^{t_a}$ to $X_j^{t_b}$ for some $t_a < t_b$, and an optional arrow from $X_i$ to itself for all $i \in \{1, \cdots, n\}$, which is also called a self-loop arrow.*

Inspired by the application of acyclic SCG (Peters et al., 2013; Assaad et al., 2022a; Wu et al., 2024), in analyzing DAGs, we adopt a holistic perspective by abstracting away from time-slice-specific causal structures and characterizing the unified causal representation (see Definition 1 and Figure (1d)) across the entire temporal sequence. For clarity in the subsequent discussion, variables with a superscript $t$ (e.g., $X_i^t$) denote nodes within a specific time slice, while variables without a superscript (e.g., $X_i$) represent their counterparts in the SCG. Our models are discussed under some constraints, including Acyclic graph, Markov property, Time-consistency, and Faithfulness.

**Assumption 1** (Acyclic summary causal graph (Assaad et al., 2022b; Wu et al., 2024)). *We assume the SCG to be acyclic, only allowing for the presence of self-loops. That is, there exists no cycle involving distinct nodes in which a node can be both an ancestor and a descendant of another.*

**Assumption 2** (Markov property (Wu et al., 2024)). *This property assumes that the future slice depends only on the current state but does not depend on its history. For example, if we assume $X_i \rightarrow X_j$, and the time lag is 1. At a random time-slice $t$, $X_j^t$ only has direct cause from $X_i^{t-1}$, there can not be any edges from $X_i^{t-\tau}$ to $X_j^t$, where $\tau > 1$.*

We assume that there are no instantaneous effects in our model; the time delay represents the minimum interval between the occurrence of causal effects. For simplicity of notation, we will not discuss the high-order lagged effects in our current model, but some analysis on that is deferred to Appendix A.1. Also, in the main text, we focus primarily on scenarios with a uniform mini-

mum time-lag $\tau$ for all causal relationships, while scenarios with heterogeneous time lags will be discussed in the Appendix A.8.

**Assumption 3** (Time-consistency (Assaad et al., 2022b; Wu et al., 2024)). *As discussed previously, the causal relationships between variables are said to be constant throughout time, which means that the causal graph has consistency on each time slice and the SCG $\mathcal{G}$.*

**Assumption 4** (Causal faithfulness (Pearl, 2009b)). *All conditional independencies in the observed data are implied by the causal graph via d-separation.*

## 2.2 Causal Interaction Structures and Backdoor Pathways

**Causal Interactions.** One of the main challenges for eliminating confoundings in causal discovery is the interaction between variables, which may lead to confusion of causal effects. Three building blocks are considered to be the foundations of causal interactions: confounder, mediator and collider effects (see Definition 2 and Figure (1a), (1b), (1c)). These interactions may be manifested individually or intertwined at the same time, leading to more complex causal graphs.

**Definition 2** (Three building blocks). *Confounder effects ($A \leftarrow B \rightarrow C$): Confounder effects are also called common cause effects: $B \rightarrow A$ and $B \rightarrow C$. This structure may cause the creation of a statistical correlation between $A$ and $C$ even if there is no arrow between them.*

*Mediator effects ($A \rightarrow B \rightarrow C$): This structure emerges in the causal chain between the variable $A$ and the variable $C$, with the variable $B$ acting as a bridge. A direct arrow from $A$ to $C$ may be created mistakenly due to the causal chain.*

*Collider effects ($A \rightarrow B \leftarrow C$): This means multiple factors acting on the same variable: $A \rightarrow B$ and $C \rightarrow B$. In this case, the variable $A$ and the variable $C$ will not be correlated unless conditioned on B.*

More discussion and examples of these blocks are provided in Appendix A.2.

**Backdoor Pathways.** When studying the causal relationship of $T$ to $Y$, we call a path $l$ from $T$ to $Y$ a backdoor pathway iff $l$ satisfies two conditions:

    (a) It contains an incoming edge to T.

    (b) It is not blocked (There is no collider).

Although the standard Backdoor Criterion (Pearl, 2009b) effectively eliminates confounders in non-time series data, it may fail when additional interactions (see Definition 2) exist. To address this, we propose an efficient algorithm that uses the backdoor pathways generated from time-lagged causation to identify true causal graphs in complex structures on time-series data.

## 3 Algorithm

In section 3.1, we will first introduce the Time-Lagged Backdoor Pathway Principle and show how the backdoor pathway is generated from the variables in time lags and how causality is discovered. Then, in section 3.2, based on the Time-Lagged Backdoor Pathway Principle and causal information flow, we propose a two-stage algorithm for causality discovery with time lags, which is applicable to different types of noise. At the last part in section 3.2, how the efects of different causal interactions are excluded will be discussed in detail.

## 3.1 Time-Lagged Backdoor Pathway Principle

**Definition 3** (Causal backdoor pathway & Non-causal backdoor pathway). *Given the time-series data $X = \{X^{t-\tau}, \cdots, X^t\}$ satisfying the definitions and assumptions before, where $\tau$ is the minimum interval between the occurrence of causal effects, if $X_i^{t-\tau} \rightarrow X_j^t$, then $X_i^t \leftarrow X_i^{t-\tau} \rightarrow X_j^t$ is a causal backdoor pathway on $X_i^t$ and $X_j^t$ since the backdoor pathway contains both self-causation from $X_i^{t-\tau} \rightarrow X_i^t$ and causation from $X_i^{t-\tau} \rightarrow X_j^t$. While if $X_k^{t-\tau} \rightarrow X_i^t$ and $X_k^{t-\tau} \rightarrow X_j^t$, where $k$ is a variable distinct from $i$ and $j$, then $X_i^t \leftarrow X_k^{t-\tau} \rightarrow X_j^t$ is a non-causal backdoor pathway on $X_i^t$ and $X_j^t$, which will lead to a confounding.*

**Theorem 1** (Time-Lagged Backdoor Pathway Principle for CIT). *For variables $X_i$ and $X_j$ in time-series data $X$, we can conclude that $X_i^{t-\tau}$ is an ancestor of $X_j^t$ iff $X_i^t \not\perp\!\!\!\perp X_j^t \mid \mathbf{C}_{X_i}^{t-\tau}$, where $\mathbf{C}_{X_i}^{t-\tau}$ is the conditioning set for $X_i$ on time-slice $t - \tau$.*

*Proof.* From the definitions and the assumptions, we can infer that:

   (a) There is a directed arrow from $X_i^{t-\tau}$ to $X_i^t$.

   (b) There is no arrow between $X_i^t$ and $X_j^t$.

   (c) Under the Markov property, $X_i^{t_a} \not\rightarrow X_i^t$ for $t_a < t - \tau < t$.

   (d) Under the acyclic assumption, $X_i^t \not\rightarrow X^{t-\tau}$.

   (e) Under the Time-consistency assumption, if $X_i \rightarrow X_j$, then $X_i^{t-\tau} \rightarrow X_j^t$ for any $t \geq \tau$.

Based on these conditions, we can infer that the correlation between $X_i^t$ and $X_j^t$ is generated from the backdoor pathways. Only causal backdoor pathway (see Definition 3) implies the existence of a causal relationship. Hence, if we cut off all the non-causal backdoor pathways (see Definition 3) by controlling the $\mathbf{C}_{X_i}^{t-\tau}$, the confounding effects on $X_i^t$ and $X_j^t$ will be eliminated. Under this circumstance, $X_i^t \perp\!\!\!\perp X_j^t \mid \mathbf{C}_{X_i}^{t-\tau}$ if $X_i^{t-\tau} \not\rightarrow X_j^t$. In turn, given the condition that $X_i^{t-\tau} \rightarrow X_j^t$, then $X_i^t \not\perp\!\!\!\perp X_j^t \mid \mathbf{C}_{X_i}^{t-\tau}$. $\qquad\square$

Having Theorem 1, the main problem turns to how to find the conditioning set $\mathbf{C}_{X_i}^{t-\tau}$. Actually, all variables at time $t - \tau$ except for $X_i^{t-\tau}$ and variables that are independent of $X_i^{t-\tau}$ can be selected into the conditioning set (Wu et al., 2024). The proof of the reasonableness for the selection of conditioning sets will be shown in Appendix B.1. We denote the conditioning set as $X_{i*}^{t-\tau}$. Thus we can reformulate our Theorem 1 using the conditioning set $X_{i*}^{t-\tau}$.

**Corollary 1.** *Given the time-series data $X = \{X^{t-\tau}, \cdots, X^t\}$, for variables $X_i$ and $X_j$, we can conclude that $X_i$ is an ancestor of $X_j$ iff $X_i^t \not\perp\!\!\!\perp X_j^t \mid X_{i*}^{t-\tau}$.*

We can use Corollary 1 to discover causality by performing a CIT on each pair of variables. Figure (1d) shows an example for the calculation process. The differences between the corollary and the traditional CIT will be analyzed in Appendix A.3. Additionally, a more detailed discussion of the comparison between Time-Lagged Backdoor Pathway Principle and the standard Backdoor Criterion is provided in the Appendix A.9. If there are enough data for interventions, we can extend our method to intervention situations (see Appendix A.4).

## 3.2 CIT-TBP Algorithm

### 3.2.1 Time-Lagged Backdoor Pathway Principle for Preliminary Causal Discovery

Based on the Time-Lagged Backdoor Pathway Principle in Corollary 1, we can discover the time-lagged causal relationships by conducting CITs on given variables. For a given dataset $X = \{X_1, \cdots, X_n\}$, we apply Hilbert-Schmidt Independence Criterion (HSIC) test (Zhang et al., 2012) with Gaussian kernel on $X_i^{t-\tau}$ and $X_j^t$, and then calculate the conditional independence significance $p$-value. If the $p$-value is less than or equal to the threshold, $X_i$ is regarded as an ancestor of $X_j$, which means $X_i^{t-\tau} \not\perp\!\!\!\perp X_j^t \mid X_{i*}^{t-\tau}$. Thus we can obtain the adjacency matrix which implies the causal graph via

$$P = \begin{bmatrix} p_{1,1} & p_{1,2} & \cdots & p_{1,n} \\ p_{2,1} & p_{2,2} & \cdots & p_{2,n} \\ \vdots & \vdots & \ddots & \vdots \\ p_{n,1} & p_{n,2} & \cdots & p_{n,n} \end{bmatrix}, \tag{1}$$

where $p_{i,j} = HSIC(X_i^{t-\tau}, X_j^t \mid X_{i*}^{t-\tau})$. $\tau$ is the minimum time lag between the occurrence of causal effects, which depends on the granularity of the observations and can be set to any reasonable positive integer. For simplicity, we set $\tau$ to 1 as the default. For each $p_{i,j}$ in $P$, if the value is less than threshold $\alpha$, the result is considered significant and the value of the corresponding position of

the adjacency matrix will be set to 1, which means there is a directed edge from node $i$ to node $j$ in the causal graph. While values of other positions in the adjacency matrix will be set to 0. $\alpha$ is typically set to 0.05, 0.01, or 0.001 in statistical hypothesis testing. In this work, we set $\alpha = 0.05$ in the main experiments. More discussion on the settings of hyperparameters is in Appendix A.5.

### 3.2.2 INFORMATION ENTROPY FOR LATENT FACTOR EXCLUSION

The results of CITs may be affected by some latent variables (Chen et al., 2024; Kivva et al., 2021). In this paper, we incorporate the concept of information entropy (Lozano-Durán & Arranz, 2022) to impose stronger constraints, thus excluding the influence of possible latent factors. Specifically, we calculate the conditional transfer entropy (CTE) between variables on different time slices and determine whether there are additional backdoor pathways in the independence test process based on the change in entropy value, so as to exclude the effects from the latent variables.

If $X_i \rightarrow X_j$, the transfer entropy (TE) (Vicente et al., 2011; Montalto et al., 2014) from $X_i$ to $X_j$ will be significantly larger than zero. Similarly, to avoid the effects of latent factors, here we introduce $\triangle CTE$. If $X_i$ and $X_j$ represent two stochastic processes, the equation for $\triangle CTE$ can be expressed as:

$$\triangle CTE_{X_i \rightarrow X_j | X_{i*}^{t-\tau}} = CTE_{X_i^{t-\tau} \rightarrow X_j^t | X_{i*}^{t-\tau}} - CTE_{X_j^t \rightarrow X_i^{t-\tau} | X_{i*}^{t-\tau}} \tag{2}$$

The detailed calculation process for CTE is in Appendix A.6. For every $i, j \in \{1, 2, \cdots, n\}$, we iteratively calculate $\triangle CTE$ for each pair of variables, if $\triangle CTE_{X_i \rightarrow X_j | X_{i*}^{t-\tau}}$ is significantly greater than 0, then we can infer that there is transfer of information from $X_i$ to $X_j$.

**Theorem 2** (CTE for the exclusion of latent factors). *Given the time-series data $X = \{X^{t-\tau}, \cdots, X^t\}$ satisfying the Definitions and Assumptions before, where $\tau$ is the minimum interval between the occurrence of causal effects, for variables $X_i$ and $X_j$, if $X_i^t \not\perp\!\!\!\perp X_j^t \mid X_{i*}^{t-\tau}$ but $\triangle CTE_{X_i \rightarrow X_j | X_{i*}^{t-\tau}}$ is not significantly greater than zero, then there is a latent factor which has effects on both $X_i$ and $X_j$.*

Proof for Theorem 2 will be presented in Appendix B.2. More discussion on the choice of significance level and the statistical robustness will be showed in Appendix A.5. Based on Theorem 2, we prune the causal graph obtained from CITs. Then, we update the values in the adjacency matrix accordingly to obtain the new matrix $A$:

$$A = \begin{bmatrix} \mathbb{I}(p_{1,1} \leq \alpha) & \cdots & \mathbb{I}(p_{1,n} \leq \alpha \ \& \ \Delta CTE_{1,n} > 0) \\ \vdots & \ddots & \vdots \\ \mathbb{I}(p_{n,1} \leq \alpha \ \& \ \Delta CTE_{n,1} > 0) & \cdots & \mathbb{I}(p_{n,n} \leq \alpha) \end{bmatrix}, \tag{3}$$

where $\mathbb{I}(\cdot)$ is an indicator function. Since the variables have no transfer entropy to themselves, the main diagonal of the adjacency matrix is slightly different. Moreover, we discuss causal discovery in the context of other CIT and CTE cases in Appendix A.7. The detailed pseudo-code for CIT-TBP is provided in Appendix C.1.

### 3.2.3 THE IDENTIFICATION OF THREE STRUCTURES.

Based on Theorem 1 and Theorem 2, we show here how our algorithm effectively excludes bias of three building blocks.

**Confounder Effects.** As shown in Figure (1a), if $X_a \leftarrow X_b \rightarrow X_c$, $X_b$ acts as a confounder when testing the causal relationship between $X_a$ and $X_c$. Through CITs on time-lagged backdoor pathways, $X_b^{t-\tau} \in (X_{a*}^{t-\tau} \cap X_{c*}^{t-\tau})$, thus $X_a \rightarrow X_c$ if $X_a^{t-\tau} \not\perp\!\!\!\perp X_c^t \mid X_{a*}^{t-\tau}$, and vice versa.

**Mediator Effects.** See Figure (1b). $X_b$ is a mediator variable between $X_a$ and $X_c$. $X_a^{t-\tau} \rightarrow X_b^t$ and $X_b^{t-\tau} \rightarrow X_c^t$ connect the backdoor pathway between $X_a^t$ and $X_c^t$. Similarly, if we conduct an HSIC test on $X_a^t$ and $X_c^t$, the backdoor pathway will be blocked.

**Collider Effects.** See Figure (1c). Our treatment variables are at time-slice $t$, while the conditioning sets are conducted at time-slice $t-\tau$. When we conduct an HSIC test, there will be no collider effects generated at time-slice $t$ and thus no spurious correlation.

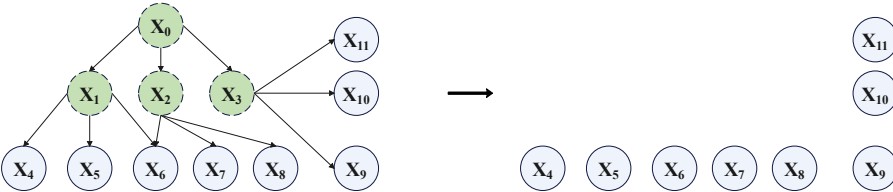

Figure 2: Causal graphs with latent factors. Green dashed nodes: latent factors. Blue solid nodes: observed variables.

Table 1: Results on different cases.

(a) Linear results.

| Algorithm | Case1 | | | Case2 | | | Case3 | | | Case4 | | |
|---|---|---|---|---|---|---|---|---|---|---|---|---|
| | SHD (↓) | F1 (↑) | FDR (↓) | SHD (↓) | F1 (↑) | FDR (↓) | SHD (↓) | F1 (↑) | FDR (↓) | SHD (↓) | F1 (↑) | FDR (↓) |
| GSP | 5.0 | 0.810 | 0.227 | 9.0 | 0.684 | 0.316 | 10.0 | 0.667 | 0.385 | 18.0 | 0.524 | 0.476 |
| GC | 13.0 | 0.727 | 0.429 | 14.0 | 0.667 | 0.448 | 22.0 | 0.638 | 0.532 | 26.0 | 0.448 | 0.649 |
| CUTS+ | 5.0 | 0.869 | 0.200 | 7.0 | 0.791 | 0.292 | 5.0 | 0.898 | 0.185 | 18.0 | 0.500 | 0.556 |
| GOLEM | 10.0 | 0.750 | 0.357 | 18.0 | 0.596 | 0.500 | 10.0 | 0.720 | 0.357 | 24.0 | 0.408 | 0.643 |
| RHINO | 5.0 | 0.869 | 0.200 | 8.0 | 0.769 | 0.250 | 6.0 | 0.880 | 0.214 | 28.0 | 0.441 | 0.658 |
| DAGMA | 6.0 | 0.800 | 0.200 | 8.0 | 0.769 | 0.250 | 8.0 | 0.714 | 0.250 | 18.0 | 0.488 | 0.500 |
| SCORE | 13.5 | 0.653 | 0.391 | 7.3 | 0.645 | 0.390 | 8.0 | 0.711 | 0.304 | 19.0 | 0.489 | 0.542 |
| PO-LINGAM | 10.0 | 0.737 | 0.222 | 12.0 | 0.632 | 0.368 | 5.0 | 0.878 | 0.053 | 16.0 | 0.550 | 0.421 |
| LPCMCI+ | 8.0 | 0.833 | 0.286 | 10.0 | 0.732 | 0.318 | 4.0 | 0.917 | 0.154 | 19.0 | 0.512 | 0.500 |
| Ours | 4.0 | 0.872 | 0.105 | 10.0 | 0.703 | 0.278 | 4.0 | 0.905 | 0.050 | 16.0 | 0.550 | 0.421 |

(b) Non-linear results.

| Algorithm | Case1 | | | Case2 | | | Case3 | | | Case4 | | |
|---|---|---|---|---|---|---|---|---|---|---|---|---|
| | SHD (↓) | F1 (↑) | FDR (↓) | SHD (↓) | F1 (↑) | FDR (↓) | SHD (↓) | F1 (↑) | FDR (↓) | SHD (↓) | F1 (↑) | FDR (↓) |
| GSP | 18.0 | 0.593 | 0.529 | 14.0 | 0.694 | 0.433 | 35.0 | 0.418 | 0.689 | 32.0 | 0.444 | 0.667 |
| GC | 21.0 | 0.588 | 0.583 | 15.0 | 0.667 | 0.449 | 38.0 | 0.376 | 0.746 | 33.0 | 0.410 | 0.726 |
| CUTS+ | 31.0 | 0.414 | 0.684 | 31.0 | 0.386 | 0.711 | 27.0 | 0.456 | 0.629 | 26.0 | 0.464 | 0.629 |
| GOLEM | 20.0 | 0.500 | 0.571 | 19.0 | 0.511 | 0.571 | 22.0 | 0.480 | 0.571 | 20.0 | 0.490 | 0.571 |
| RHINO | 21.0 | 0.528 | 0.576 | 26.0 | 0.462 | 0.674 | 28.0 | 0.424 | 0.682 | 26.0 | 0.452 | 0.659 |
| DAGMA | 21.0 | 0.500 | 0.542 | 17.0 | 0.558 | 0.500 | 22.0 | 0.511 | 0.520 | 21.0 | 0.489 | 0.542 |
| SCORE | 13.2 | 0.594 | 0.353 | 13.6 | 0.611 | 0.353 | 16.5 | 0.564 | 0.353 | 13.4 | 0.629 | 0.214 |
| PO-LINGAM | 18.0 | 0.550 | 0.450 | 14.0 | 0.600 | 0.429 | 18.0 | 0.537 | 0.421 | 16.0 | 0.636 | 0.391 |
| LPCMCI+ | 1.0 | 0.976 | 0.047 | 8.0 | 0.780 | 0.272 | 23.0 | 0.480 | 0.571 | 24.0 | 0.480 | 0.586 |
| Ours | 10.0 | 0.667 | 0.000 | 9.0 | 0.690 | 0.000 | 12.0 | 0.625 | 0.000 | 11.0 | 0.645 | 0.000 |

**Bold** indicates the best performance, underline indicates the second-best.

## 4 EXPERIMENTS

### 4.1 BASELINES AND EVALUATION.

In these experiments, we provide various kinds of causal discovery baseline methods, using their time series variants in our tasks. We test the proposed CIT-TBP to both synthetic and real-world data and compare its performance to the following baselines: time-series method, GC (Barnett & Seth, 2014), CUTS+ (Cheng et al., 2024a), RHINO (Gong et al., 2023); constraint-based methods, FCI+ (Entner & Hoyer, 2010; Malinsky & Spirtes, 2018),GSP (Solus et al., 2021), LPCMCI+ (Günther et al., 2023), PO-LINGAM (Jin et al., 2024); score-based Methods, GOLEM (Ng et al., 2020), DAGMA (Bello et al., 2022); topology-based method, SCORE (Rolland et al., 2022).

To evaluate the performance of our proposed CIT-TBP method, we employ three standard metrics: the Structural Hamming Distance (**SHD**) to measure overall structural differences, the **F1-Score** to

assess edge identification accuracy, and the False Discovery Rate (**FDR**) to quantify the proportion of falsely discovered edges, which is also important as false causal claims often have more serious practical consequences than missed discoveries. Discussions on the evaluation metrics and implementation specifics are provided in Appendix C.5. Details of computations are in Appendix C.2.

## 4.2 EXPERIMENTS ON SYNTHETIC DATA

**Synthetic Data.** We tested our CIT-TBP on synthetic data generated from SCMs under Assumptions 1, 2 and 3. In three-block exclusion experiments and latent factor exclusion experiments, we manually set up causal graphs for different cases. In other experiments, we generated causal graphs using the Erdos-Renyi model (Erdös & Rényi, 2006). In our main experiments, we generated the data with Gaussian Noise for every variable $X_i^h$, where variables $i = 1, 2, \cdots, n$ and time $h = 1, 2, \cdots, 1000$. We evaluated our algorithms on both linear and non-linear structural equations:

$$X_j^t = \frac{3}{5} \times X_j^{t-\tau} + \sum_{X_i^{t-\tau} \in Pa(X_j^t) \setminus X_j^{t-\tau}} (\frac{1}{5} \times X_i^{t-\tau}) + \epsilon_j^t, \tag{4}$$

$$X_j^t = \frac{5}{1 + exp(-X_j^{t-\tau})} + \sum_{X_i^{t-\tau} \in Pa(X_j^t) \setminus X_j^{t-\tau}} (\frac{1}{1 + exp(-X_i^{t-\tau})}) + \epsilon_j^t,$$

$$X^0 \sim \mathcal{N}(0, 1), \epsilon^t \sim \mathcal{N}(0, 0.4), \tag{5}$$

**Experiments Setup.** We evaluated the performance of our CIT-TBP on three building blocks (See Figure 1). Then, we varied the number of nodes and edges to see the performance on causal graphs of different sizes and different densities. For robustness tests, we generated data with different kinds of noise for experiments. In addition, we constructed causal graphs containing latent factors and tested the ability of our CIT-TBP to exclude latent factors. The details of the experiment setup are provided in Appendix C.3.

Table 2: Robustness tests under different noise types.

(a) Linear causality.

| Algorithm | Gaussian | | | Laplace | | | Uniform | | |
|---|---|---|---|---|---|---|---|---|---|
| | SHD($\downarrow$) | F1($\uparrow$) | FDR($\downarrow$) | SHD($\downarrow$) | F1($\uparrow$) | FDR($\downarrow$) | SHD($\downarrow$) | F1($\uparrow$) | FDR($\downarrow$) |
| RHINO | 8.0 | 0.640 | 0.333 | 8.0 | 0.615 | 0.333 | 8.0 | 0.640 | 0.333 |
| DAGMA | 8.0 | 0.733 | 0.267 | 8.0 | 0.733 | 0.267 | 8.0 | 0.733 | 0.267 |
| SCORE | 8.0 | 0.733 | 0.267 | 10.4 | 0.645 | 0.375 | 8.0 | 0.733 | 0.267 |
| PO-LINGAM | 9.0 | 0.710 | 0.313 | 8.0 | 0.710 | 0.313 | **6.0** | **0.800** | **0.200** |
| Ours | **6.0** | **0.786** | **0.154** | **7.0** | **0.759** | **0.214** | 7.0 | 0.759 | 0.214 |

(b) Non-linear causality.

| Algorithm | Gaussian | | | Laplace | | | Uniform | | |
|---|---|---|---|---|---|---|---|---|---|
| | SHD($\downarrow$) | F1($\uparrow$) | FDR($\downarrow$) | SHD($\downarrow$) | F1($\uparrow$) | FDR($\downarrow$) | SHD($\downarrow$) | F1($\uparrow$) | FDR($\downarrow$) |
| RHINO | **8.0** | 0.640 | 0.333 | 7.0 | 0.720 | 0.250 | 11.0 | 0.552 | 0.333 |
| DAGMA | 10.0 | 0.688 | 0.353 | 12.0 | 0.647 | 0.421 | 15.0 | 0.595 | 0.500 |
| SCORE | 11.2 | 0.667 | 0.389 | 13.4 | 0.629 | 0.450 | 13.3 | 0.588 | 0.474 |
| PO-LINGAM | **8.0** | **0.750** | 0.294 | 10.0 | 0.645 | 0.375 | 9.0 | 0.710 | 0.313 |
| Ours | **8.0** | 0.714 | **0.231** | **6.0** | **0.786** | **0.154** | **7.0** | **0.759** | **0.214** |

**Bold** indicates the best performance, underline indicates the second-best.

**Experiments on Different Structures.** We tested our CIT-TBP on three building blocks and their hybrid structure: i) confounders, ii) Colliders, iii) Mediators, iv) Hybrids. From the results in Table 1, our approach shows excellent performance in most cases, especially in the hybrids. We also designed two additional functions to test our algorithm. The causal graphs of the four cases and the additional experiments are deferred to Appendix D.1.

Table 3: The results on latent factor exclusion.

| Methods | Linear | | Non-linear | |
|---|---|---|---|---|
| | SHD ($\downarrow$) | Error-Score ($\downarrow$) | SHD ($\downarrow$) | Error-Score ($\downarrow$) |
| FCI | 1.0 | 1.0 | 4.0 | 4.0 |
| LPCMCI+ | 5.0 | 5.0 | 17.0 | 17.0 |
| PO-LINGAM | 7.0 | 7.0 | 8.0 | 8.0 |
| Ours | **0.0** | **0.0** | **0.0** | **0.0** |

**Bold** indicates the best performance.

**Robustness Experiments.** We additionally generated data with Laplace and Uniform noise. Here we chose excellent methods in each kind of causal algorithms in the former experiments for comparison. The results are presented in Table 2, which demonstrate the robust and superior performance of our algorithms against different types of noise. What's more, experiments on varying different numbers of nodes and edges are deferred to Appendix D.2.

**Experiments on Latent Variables Exclusion.** We constructed the causal graph manually and removed key nodes in the data input process (see Figure 2) to simulate the presence of latent factors. Methods that do not rely on the assumption of no latent factors are selected for comparison. We evaluate the performance through SHD and Error-Score (ES), where ES is used to calculate how many wrong edges caused by latent factors are identified. We found that our approach accurately excludes the influence of the latent factors (see Table 3). To enhance generalizability, we further validated our findings on two additional causal graphs under different settings (Appendix D.3).

## 4.3 EXPERIMENTS ON REAL-WORLD DATA.

**CausalTime** (Cheng et al., 2024b) is a novel pipeline capable of generating realistic time-series data. We used three types of benchmark time series with time-lagged causality from weather, traffic, and healthcare backgrounds. There are 20 nodes in the Traffic and Medical Dataset, and 36 nodes in the AQI Dataset. We compared the ability of our method with baseline methods in recovering the true causal graph. From the results in Table 4, our algorithm identifies the most accurate causal graph under all realistic datasets. More discussions will be shown in Appendix C.4.

Table 4: Causal discovery on CausalTime dataset.

| Methods | SHD ($\downarrow$) | | |
|---|---|---|---|
| | Traffic | Medical | AQI |
| RHINO | $63.248 \pm 1.732$ | $80.202 \pm 2.236$ | $165.945 \pm 6.016$ |
| DAGMA | $59.790 \pm 3.213$ | $91.456 \pm 4.115$ | $244.713 \pm 9.980$ |
| SCORE | $66.298 \pm 7.980$ | $100.350 \pm 8.419$ | $280.823 \pm 32.564$ |
| PO-LINGAM | $49.498 \pm 1.835$ | $88.085 \pm 3.128$ | $179.327 \pm 3.515$ |
| Ours | $\mathbf{31.346 \pm 1.033}$ | $\mathbf{75.192 \pm 0.893}$ | $\mathbf{160.062 \pm 2.597}$ |

**Bold** indicates the best performance.

## 5 CONCLUSIONS AND FUTURE WORKS

In this paper, we propose a causal discovery algorithm CIT-TBP to exclude different kinds of confounding effects in time-lagged causal relationships, without prior knowledge regarding the presence of latent variables. Furthermore, we theoretically demonstrated the effectiveness of CIT-TBP. We have reduced the search space of the conditioning sets compared to traditional constraint-based methods, but CITs still have limitations when dealing with large-scale graphs. One future research is towards reducing the number of CITs on large and dense graphs. Another direction for future research is the discovery of causal graphs in high-dimensional data.

## 6 ETHICS STATEMENT

This paper presents work whose goal is to advance the field of Machine Learning and Causal Discovery. The research conducted in the paper conform, in every respect, with the ICLR Code of Ethics.

## 7 REPRODUCIBILITY STATEMENT

All experiments in this paper is reproducible, experimental setup and complete experimental results are depicted in Section 4 and Appendix D. Our code is available at `https://anonymous.4open.science/r/CIT-TBP-F0E8`.

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

# A ADDITIONAL DISCUSSIONS

## A.1 DISCUSSIONS ON HIGH-ORDER LAGGED EFFECTS

We can also relax the Markov Assumption to a high-order Markov Assumption (Wu et al., 2024; Peters et al., 2017). In high-order Markovian hypothesis, the causal effects are on multiple time delays. In this case, $X^t$ depends on states $X^{t-a,\cdots,t-\tau}$, where $a$ is an integer greater than 0. The conditioning set is not set on a single time slice $t - \tau$, but on multiple time slices $t - a, \cdots, t - \tau$. Then, the conditioning sets can be replaced from $X_{i*}^{t-\tau}$ to $X_{i*}^{t-a,\cdots,t-\tau}$. For example, if $a = 3$ and $\tau = 1$, when we explore the causal relationships between $X_i^t$ and $X_j^t$, the conditioning set is $\{X_{i*}^{t-3}, X_{i*}^{t-2}, X_{i*}^{t-1}\}$.

## A.2 DIFFERENT STRUCTURES AND EXAMPLES

**Confounders.** Confounding factors are the most common interactions between nodes. One of the examples is the relationship between education level and income. While education level $(A)$ and income $(C)$ show a strong correlation, the apparent causal relationship $A \to C$ may be spurious. The underlying driver is actually family socioeconomic status $(B)$, which influences both educational opportunities and career advantages.

**Mediators.** When examining neighborhood security $(A)$, residents' trust $(B)$, and community engagement $(C)$, researchers may overlook critical causal pathways. Improved security enhances trust in the community, which in turn increases participation in local activities. However, if analysts observe only the surface correlation between $A$ and $C$, they might incorrectly conclude that "better security directly boosts engagement", neglecting trust's essential mediating role. This oversimplification leads to flawed policy interventions, like allocating resources solely to policing while ignoring trust-building programs, that fail to address the actual mechanism through which security improvements ultimately affect community participation.

**Colliders.** Collider bias can induce spurious associations when conditioning on a common effect. Consider the relationship between single-parent households $(A)$, mental health issues $(B)$, and academic performance $(C)$. Here, $(B)$ is a collider: $(A \to B \leftarrow C)$. Single-parent households may increase psychological stress, while poor academic performance can worsen mental health. If researchers condition on $B$. (e.g., studying only individuals with mental health issues), a spurious correlation emerges between $A$ and $C$. This may lead to erroneous conclusions like "single-parenthood directly harms academic performance," even though no such causal link exists.

## A.3 THE DIFFERENCE BETWEEN TRADITIONAL CONDITIONAL INDEPENDENCE TESTS AND OURS

We compare our method with the traditional constraints-based method PC Algorithm in the selection of the conditioning sets. The PC algorithm is shown in Algorithm 1.

1. In PC algorithm, the conditioning sets are dynamically constructed, depending on the neighbor relationships of the graph and the current structure during iteration. However, the conditioning sets are statically constructed without iteration, which can reduce time complexity.

2. PC algorithm is only suitable for data without a temporal order. Runge et al. (Runge et al., 2019) proposed an improved version of the PC algorithm, PCMCI, which adapts it for time-series data. However, the conditional independence tests used in these methods do not adequately account for time-lagged causal relationships between variables, leaving them vulnerable to confounding structures and still requiring the assumption of no latent variables. Our algorithm use time-lagged backdoor paths to eliminate the effects of different structures. Unlike traditional methods, we control conditioning sets and treatment variables in different time-slice layers, which can remove the effects of V-structure.

3. We also introduce the concept of CTE and evaluate the difference in entropy between different layers to determine the presence of potential factors, thereby relaxing the constraints imposed by traditional algorithms on latent variables.

---

**Algorithm 1:** The PC-pop Algorithm

---

**Data:** Vertex Set $V$; Condition Independence Information
**Result:** Causal Graph $C$; separation sets $S$

1   Form the complete undirected graph $\tilde{C}$ on the vertex set $V$.
2   $\ell = -1; C = \tilde{C}$
3   **repeat**
4     $\ell = \ell + 1$
5    **repeat**
6      Select a (new) ordered pair of nodes $i, j$ that are adjacent in $C$ such that $|\text{adj}(C, i) \setminus \{j\}| \geq \ell$;
7      **repeat**
8       Choose (new) $k \subseteq \text{adj}(C, i) \setminus \{j\}$ with $|k| = \ell$;
9       **if** $i$ and $j$ are conditionally independent given $k$, **then**
10        Delete edge $i, j$;
11        Denote this new graph by $C$;
12        Save $k$ in $S(i, j)$ and $S(j, i)$;
13      **end**
14      **until** edge $i, j$ is deleted or all $k \subseteq \text{adj}(C, i) \setminus \{j\}$ with $|k| = \ell$ have been chosen;
15    **until** all ordered pairs of adjacent variables $i$ and $j$ such that $|\text{adj}(C, i) \setminus \{j\}| \geq \ell$ and $k \subseteq \text{adj}(C, i) \setminus \{j\}$ with $|k| = \ell$ have been tested for conditional independence;
16   **until** for each ordered pair of adjacent nodes $i, j$: $|\text{adj}(C, i) \setminus \{j\}| < \ell$.

---

Our CIT-TBP offers a significant improvement over the PC algorithm by optimizing the selection of conditioning sets. Unlike the PC algorithm, which dynamically explores and stores multiple combinations of conditioning sets, our approach uses a pre-determined conditioning set based on time layers. This eliminates the need for extensive exploration and reduces the time requirements, making it more efficient. The search space for the conditioning sets is smaller than that in the traditional constraint-based approaches (Wu et al., 2024; Solus et al., 2021; Shiragur et al., 2024). We provide $Big\text{-}O$ analysis in A.10.

### A.4   DISCUSSIONS ON INTERVENTIONS

In structural causal models (SCMs), interventions are represented by the $do(\cdot)$ operator (Pearl, 2009a; Xu et al., 2024). When a variable is intervened on, it is no longer influenced by its parent variables. As a result, the causal graph of the intervened system will no longer contain any directed edges into that variable. In our CIT-TBP, the conditioning set will be replaced by an empty set since the intervention operation makes the treatment variable independent of others. The formula of interventions is shown in Corollary 2.

**Corollary 2.** *Let $do(X_i^{t-\tau})$ be an intervention. For variables $X_i$ and $X_j$, we can conclude that $X_i$ is an ancestor of $X_j$ iff $do(X_i^{t-\tau}) \not\perp\!\!\!\perp X_j^t$.*

However, interventions are often challenging to implement in practice, as they typically require prior knowledge of the system (Li et al., 2024; Jeunen et al., 2022). One key direction for future research will be to develop more effective methods for generating high-quality intervention data.

### A.5   SETTINGS OF HYPERPARAMETER

**Hyperparameters for CIT.** The hyperparameter configuration for CIT includes time delay $\tau$, significance level $\alpha$, and transfer entropy bandwidth $k$. For time delay, we set $\tau = 1$ as default. (Alternative values are discussed in Appendix A.1.) The significance level can be adjusted contextually: 0.05 for linear/sparse cases to capture more causal relationships, and 0.01, even 0.001, for complex nonlinear scenarios to ensure precise causal relationships. For the transfer entropy bandwidth, it can be discretionary according to the distribution of the data, we use 2.0 as default.

**Significance Level and Statistical Robustness for CTE.** To quantify the uncertainty of our estimates, we computed 95% confidence intervals using the nonparametric bootstrap method (Owens

& Hekman, 2016). A standard number of $B = 5000$ replicates was used. For the setting of significance level, different significance level indeed have an impact on the results. A higher significance level means that our causal restrictions are looser, while a lower significance level means that we set more strict causal rules. We use $\alpha = 0.05$ as the default setting (95% confidence interval), which is also a value commonly used in the significance tests (Liu et al., 2023; Wu et al., 2024).Those hyperparameters of our method can be flexibly adjusted according to usage conditions.

### A.6 CALCULATION OF THE CTE

The shannon entropy is calculated as:

$$H(X) = \Sigma_i p(x) \cdot \log_2 \frac{1}{p(x)} = -\Sigma_i p(x) \cdot \log_2 p(x) \tag{6}$$

TE is a model-free, information-theoretic measure of directed, dynamic information flow between two processes. It quantifies how much the historical states of a source process $X$ reduce uncertainty about the current state of a target process $Y$, beyond what is already contained in the history of $Y$. The formula is shown below:

$$TE_{X_t \to Y_t} = H(Y_t|Y_{t-\Delta w}) - H(Y_t|Y_{t-\Delta w}, X_{t-\Delta \tau})$$

$$= H(Y_t, Y_{t-\Delta w}) - H(Y_{t-\Delta w}) - H(Y_t, Y_{t-\Delta w}, X_{t-\Delta \tau}) + H(Y_{t-\Delta w}, X_{t-\Delta \tau})$$

$$= H(Y_t, Y_{t-\Delta w}) + H(Y_{t-\Delta w}, X_{t-\Delta \tau}) - H(Y_{t-\Delta w}) - H(Y_t, Y_{t-\Delta w}, X_{t-\Delta \tau}) \tag{7}$$

$$= \sum_{x_\tau, y, y_w} p(x_\tau, y, y_w) \log_2 \frac{p(x_\tau, y, y_w)p(y_w)}{p(x_\tau, y_w)p(y, y_w)}$$

CTE is a generalization of transfer entropy that quantifies the direct information transfer from a source process $X$ to a target process $Y$, conditional on one or more additional processes $Z$. It isolates the unique causal influence of $X$ on $Y$ by accounting for:

1) Confounding effects ($X \leftarrow Z \to Y$).
2) Mediated effects ($X \to Z \to Y$).

Thus, the formula for CTE can be derived from the TE:

$$CTE_{X_t \to Y_t|Z_t} = H(Y_t \mid Y_{t-\Delta w}, Z_{t-\Delta w}) - H(Y_t \mid Y_{t-\Delta w}, Z_{t-\Delta w}, X_{t-\Delta \tau})$$

$$= \sum_{x_\tau, y, y_w, z_w} p(x_\tau, y, y_w, z_w) \log_2 \frac{p(x_\tau, y, y_w, z_w)p(y_w, z_w)}{p(x_\tau, y_w, z_w)p(y, y_w, z_w)} \tag{8}$$

### A.7 DISCUSSION ON DIFFERENT CONDITIONS ON CITs AND CTE

In the main text, we determine causal relationships between variables based on two criteria: the $p$-value being less than or equal to the threshold $\alpha$, and the difference of CTE being significantly greater than 0. We demonstrate that when the $p$-value is below $\alpha$ but the CTE difference is not significantly greater than 0, latent factors may exist in the system, leading to spurious correlations. Here, we further examine the remaining two scenarios.

1) $p$-value $\geq \alpha$ and $\Delta CTE > 0$: There are two possible reasons for this phenomenon. a.) First, there may exist a strongly nonlinear causal relationship between the two variables, which makes it difficult to detect the dependency using simple correlation measures. For example, in certain dynamic systems, some signal pathways may exhibit no significant correlation but still carry causal influence. b.) It may result from the issue of causal granularity (Foroni & Marcellino, 2015). The data we observe may be at a relatively coarse temporal resolution, which makes it hard to accurately capture the causal transmission. The actual

causal effect may have occurred between two observable time points, leading to observable changes in information entropy, while the correlation remains low due to the temporal mismatch.

2) $p$-value $\geq \alpha$ and $\Delta CTE = 0$: In this case, we consider that there is no causal relationship at all between the two variables.

Our discussion builds on the ideal case where there is no complex noise interference.

### A.8 CAUSAL RELATIONSHIPS WITH HETEROGENEOUS TIME LAGS

Here, we extend the algorithms to a scenario with heterogeneous lags, where for instance $X_1 \to X_2$ has a lag of $\tau_1 = 5$ and $X_3 \to X_4$ has a lag of $\tau_2 = 1$. In this case, we manually set the maximum time delay $\tau_{max}$ and then choose all variables that are correlated with the treatment variable from $t - \tau_{max}$ to $t - \tau$ into the conditioning set. In this way, the non-causal backdoor paths with the highest time delay in a heterogeneous lags structure can be directly blocked, and the non-causal backdoor paths of each time slice between variables with lower time delays can also be blocked, thereby achieving reliable causal identification. For example, if $\tau_{max} = 5$ and $\tau = 1$, we will choose all variables that are correlated with the treatment variable from $t - 5$ to $t - 1$ into the conditioning set and block them.

### A.9 THE COMPARISON BETWEEN TIME-LAGGED BACKDOOR PATHWAY PRINCIPLE AND THE STANDARD BACKDOOR CRITERION

The standard Backdoor Criterion only proposes how to establish a valid conditioning set. We make sufficient use of the concept of backdoor pathways introduced in the backdoor criterion, retain the causal backdoor pathways, and exclude non-causal backdoor pathways to perform causal discovery on time-series data (see Definition3). Therefore, in Theorem 1, we provide a novel calculation method called "Time-Lagged Backdoor Pathway Principle".

More importantly, The standard Backdoor Criterion only handle confounders and it is mainly used in non-temporal data. Our Time-Lagged Backdoor Pathway Principle is designed for various complex causal structures (including colliders, mediators, and their hybrids) on time-series data.

### A.10 $Big\text{-}O$ ANALYSIS BETWEEN CIT-TBP AND PC-BASED METHODS

**PC-based methods:** For a pair $(i, j)$, the PC-algorithm needs to enumerate all possible conditional set sizes $s = 0, 1, \ldots, d_{max}$, and select $\begin{pmatrix} d_{max} \\ s \end{pmatrix}$ subsets from the neighbor set for conditional independence tests, where $d_{max}$ represents the maximum number of neighbors (degree). If the number of node is $p$, the worst case for CIT is: $T_{CIT_{PC}} = \sum_{i<j} \sum_{s=0}^{d_{max}} \begin{pmatrix} d_{max} \\ s \end{pmatrix} = O(p^2 2^{d_{max}})$.

For HSIC tests, if the sample size is $n$, $O(n^2)$ is usually required, so the overall time complexity is: $T_{PC} = O(p^2 2^{d_{max}} n^2)$.

**CIT-TBP:** For adjacency relationship $(i, j)$, only two HSIC tests are performed. If the number of edges is $E$, the time complexity is: $T_{CIT} \approx 2En^2 = O(En^2)$. In the second stage, we assume the number of edges that pass the screening is $E_p$, and calculate $CTE$ twice for each edge, with $B$ bootstrap times, for total of $2(1+B)$. The complexity of each $CTE$ calculation is about $O(n \log n)$, and the time complexity is $T_{CTE} = O(E_p(1 + B)n \log n)$. The overall complexity is: $T_{total} = O(En^2 + E_p(1 + B)n \log n)$.

In comparison, the time complexity of CIT-TBP is obviously lower.

## B PROOFS

### B.1 PROOF OF THE RATIONALITY OF CONDITIONING SETS SELECTION

*Proof.* For time-series data with time-lagged causality, the conditioning sets $X_{i*}^{t-\tau}$ are at the $t - \tau$ time-slice, while the treatment variables $X_i^t$ and outcome variables $X_j^t$ are at the $t$ time-slice. The

set of variables can be divided into three categories, $a$) variables that are not related to $X_i^{t-\tau}$, $b$) variables that are affected by $X_i^{t-2\tau}$, $c$) and variables that affect $X_i^{t-\tau}$ at time-slice $t - 2\tau$. For variables that are not related to $X_i^{t-\tau}$, we do not need to consider them. For variables $X_k^{t-\tau}$ in $b$), they may lead to Mediators at time $t$ if $X_k \to X_j$, and if $X_k \not\to X_j$, the conditioning sets do not introduce additional backdoor pathways at time-slice $t$. The proof of variables in case $c$) is similar. Therefore, the variables associated with $X_i^{t-\tau}$ are chosen as the conditioning sets. $\square$

### B.2 PROOF OF THE EXCLUSION OF LATENT FACTORS

*Proof.* We assume that $X_i$ doesn't cause $X_j$, but there is a latent factor U which affects both $X_i$ and $X_j$. When we calculate the conditional independence test on $X_i^{t-\tau}$ and $X_j^t$, the result will be significant. Actually, there is no directed edge from $X_i^{t-\tau}$ to $X_j^t$ and only the backdoor pathway $X_i^{t-\tau} \leftarrow U \to X_j^t$ contributes to the spurious association. In this case, if $\triangle CTE_{X_i \to X_j | X_{i*}^{t-\tau}}$ is not significantly greater than zero, that means there are no information transfers from $X_i^{t-\tau}$ to $X_j^t$, then the edge from $X_i^{t-\tau} \to X_j^t$ can be inferred as a mistake. A detailed mathematical proof is shown below.

We calculate the conditional transfer entropy on $X_i^{t-\tau}$ and $X_j^t$, and we use $X_z^{t-\tau}$ to represent the condition variables $X_{i*}^{t-\tau}$. The formula is like:

$$
\begin{aligned}
&\triangle CTE_{X_i^{t-\tau} \to X_j^t | X_z^{t-\tau}} \\
&= CTE_{X_i^{t-\tau} \to Y_j^t | X_z^{t-\tau}} - CTE_{Y_j^t \to X_i^{t-\tau} | X_z^{t-\tau}} \\
\\
&= H(X_j^t \mid X_j^{t-\Delta w}, X_z^{t-\Delta w}) - H(X_j^t \mid X_j^{t-\Delta w}, X_z^{t-\Delta w}, X_i^{t-\Delta \tau}) \\
\\
&\quad - H(X_i^{t-\tau} \mid X_i^{t-\Delta \tau}, X_z^{t-\Delta w}) + H(X_i^{t-\tau} \mid X_i^{t-\Delta \tau}, X_z^{t-\Delta w}, X_j^{t-\Delta w})
\end{aligned}
\tag{9}
$$

When there is a causal relationship between $X_i^{t-\tau}$ and $X_j^t$, we can get,

$$
H(X_j^t \mid X_j^{t-\Delta w}, X_z^{t-\Delta w}) - H(X_j^t \mid X_j^{t-\Delta w}, X_z^{t-\Delta w}, X_i^{t-\Delta \tau}) > 0
\tag{10}
$$

At the same time, it can be known based on established knowledge that, $X_j^{t-\Delta w} \not\to X_i^{t-\Delta \tau}$, so we conclude that,

$$
H(X_i^{t-\tau} \mid X_i^{t-\Delta \tau}, X_z^{t-\Delta w}) = H(X_i^{t-\tau} \mid X_i^{t-\Delta \tau}, X_z^{t-\Delta w}, X_j^{t-\Delta w})
\tag{11}
$$

Combining Equation (11) and Equation (12), it is easily to know that the formula in Equation (10) should be significantly greater than 0.

On the other hand, if there is no causal relationship between them, The spurious correlation caused by common factors lacks directionality-it injects the same amount of information into both sequences simultaneously. Therefore, when measured by mutual information, the amount of information flowing from $X_i$ to $X_j$ is equal to that from $X_j$ to $X_i$, and the difference cancels out. Under rigorous statistical tests, the bidirectional CTE is not significantly greater than zero, that is to say,

$$
\begin{aligned}
&H(X_j^t \mid X_j^{t-\Delta w}, X_z^{t-\Delta w}) - H(X_j^t \mid X_j^{t-\Delta w}, X_z^{t-\Delta w}, X_i^{t-\Delta \tau}) \\
\\
&\approx H(X_i^{t-\tau} \mid X_i^{t-\Delta \tau}, X_z^{t-\Delta w}) - H(X_i^{t-\tau} \mid X_i^{t-\Delta \tau}, X_z^{t-\Delta w}, X_j^{t-\Delta w})
\end{aligned}
\tag{12}
$$

Thus, we can conclude that under this circumstance, Equation (10) equals to 0. $\square$

A zero difference in conditional transfer entropy is a sufficient but not necessary condition for the presence of a common latent factor. However, using this condition as a constraint for pruning a preliminary causal graph can significantly reduce confounding caused by latent factors. Consequently, our method relaxes the assumption of causal sufficiency (Pearl, 2009b).

## C CIT-TBP Algorithm and Detailed Experimental Setup

### C.1 Pseudo-Code of Our CIT-TBP

Using Definitions 1 and 2, Assumptions 1, 2 and 3, we propose CIT-TBP, a novel algorithm using time-lagged causal backdoor pathways to infer causality in different complex confounding structures without the constraints on latent variables. The pseudo-code of CIT-TBP is shown in Algorithm 2.

---

**Algorithm 2:** Our CIT-TBP Algorithm

1 **Input:** Time-series data $X = \{X^0, \cdots, X^t\}$ with time-lags causation; Vertex Set $V$ with $n$ nodes.
2 **Output:** One adjacency matrix $A$.
3 **Components:** $HSIC(\cdot); CTE(\cdot)$.
4 **for** i=1 **to** $n$ **do**
5     Set the conditioning set as $X_{i*}^{t-\tau} = \{X_k^{t-\tau} \mid X_k^{t-\tau} \not\perp\!\!\!\perp X_i^{t-\tau}, k \neq i\}$
6     **for** j=1 **to** $n$ **do**
7         **if** $X_i = X_j$, **then**
8             $p_{i,j} = HSIC(X_i^{t-\tau}, X_j^t \mid X_{i*}^{t-\tau})$
9             $a_{i,j} = \mathbb{I}(p_{i,j} \leq \alpha)$
10         **elif** $X_i \neq X_j$, **then**
11             $p_{i,j} = HSIC(X_i^{t-\tau}, X_j^t \mid X_{i*}^{t-\tau})$
12             $\triangle CTE_{X_i \to X_j \mid X_{i*}^{t-\tau}} = CTE_{X_i^{t-\tau} \to X_j^t \mid X_{i*}^{t-\tau}} - CTE_{X_j^t \to X_i^{t-\tau} \mid X_{i*}^{t-\tau}}$
13             $a_{i,j} = \mathbb{I}(p_{i,j} \leq \alpha \ \& \ \triangle CTE_{i,j} > 0)$
14     **end for**
15 **end for**
16 We obtain $P = \{p_{i,j}\}_{n \times n}$ and $A = \{a_{i,j}\}_{n \times n}$.

---

### C.2 Hardware and Software Used

**Hardware used:** Ubuntu 22.04.4 LTS with 2 Œ Intel(R) Xeon(R) Platinum 8336C CPU @ 2.30GHz (64 cores, 32 cores per socket, 1 thread per core), 251 GB RAM.

**Software used:** Python 3.8, scikit-learn 1.3.2, networkx 3.1, numpy 1.23.5.

### C.3 Synthetic Data Experiments Setup

**Experiments on Different Noise.** To test the robustness of our CIT-TBP, we use synthetic data with different types of noise. The settings are:

$$Gaussian\ Noise : X^0 \sim \mathcal{N}(0,1), \epsilon^t \sim \mathcal{N}(0, 0.4) \tag{13}$$

$$Laplace\ Noise : X^0 \sim \mathcal{N}(0,1), \epsilon^t \sim Laplace(0, \frac{\sqrt{2}}{5}) \tag{14}$$

$$Uniform\ Noise : X^0 \sim \mathcal{N}(0,1), \epsilon^t \sim U(-\frac{2\sqrt{3}}{5}, \frac{2\sqrt{3}}{5}) \tag{15}$$

**Experiments on Latent Factors.** In these experiments, we constructed the causal graph manually. The causal graph is displayed in Figure 2. The causal graph consists of 12 nodes, from $X_0$ to $X_{11}$. In the data input phase, we hide $X_0$, $X_1$, $X_2$ and $X_3$ as latent variables. In this case, the true causal graph changes, and no directed edge is preserved. However, the latent variables may lead to spurious associations. For example, $X_4 \to X_5$ because of $X_1$ (Confounder). In the latent factors exclusion experiment, our goal is to ignore these spurious correlations and find the true causal relationships. We use SHD and ES to evaluate the performance of the methods. ES is incremented by 1 per spurious correlation-induced edge. The results are shown in Table 3. Besides, we also conducted experiments on more complex cases. Those results are shown in Appendix D.3.

## C.4 Real-World Data Experiments Setup

**CausalTime** (Cheng et al., 2024b) is a novel pipeline to generate realistic time series with ground truth causal graphs. In our experiments, we use 3 types of benchmark time-series from weather, traffic, and healthcare scenarios generated from the pipeline. Each dataset contains two files, $gen\_data.npy$ and $graph.npy$. Each dataset has 480 samples. We inferred causal graphs for all 480 samples, compared them with the true graph in $graph.npy$. The mean and the standard deviation of error were reported. (See Table 4).

## C.5 Experiment Details

**Discussion on the Evaluation Metrics.** To test the performance of our CIT-TBP, we compute the Structural Hamming Distance (**SHD**) between the predicted graphs and the true graphs, which measures the differences in terms of node, edge, and connection count in two graphs. The accuracy of the identified edges can also be evaluated by **F1-Score** between two graphs. In real scenarios, the downstream task of causal discovery is often interventions. If no causal edge is found, the worst case is no intervention, which will not affect the current states. If the wrong edge is inferred, it may lead to wrong intervention, which may bring worse effects and potential ethical issues. For example, in clinical medicine, if the effect of the drug on the patient is not found, it will not lead to worse results. If the causal edge is misjudged, such as using the wrong drug for intervention, it may cause the patient's condition to worsen. Therefore, we introduce an additional metric **FDR** to measure how many of the edges identified are false. However, it doesn't mean that we can remove a edge more easily when we use this indicator. FDR is just another perspective on understanding the effectiveness of algorithms, if focusing only on SHD and F1-score, our method still performs good.

**Implementation Details.** In experiments with synthetic data, algorithms without guaranteed unique topological ordering were executed 10 times with averaged results. For the fairness of the experiments comparison, we have already blacklisted the anti-causal edges derived from those algorithms that are not designed to work on time series data.

# D Supplementary Experiments

## D.1 Experiments on Different Causal Structures

In the experiments, we constructed causal structures of different interactions, including: i) confounders, ii) Colliders, iii) Mediators, iv) Hybrids. The causal graphs are shown in Figure 3.

In addition to the two structural equations mentioned in the main text, we have also constructed another set of structural equations, consisting of one linear equation and one non-linear equation. The structural equations are as follows:

$$X_j^t = \frac{1}{2} \times X_j^{t-\tau} + \sum_{X_i^{t-\tau} \in Pa(X_j^t) \setminus X_j^{t-\tau}} (\frac{2}{5} \times X_i^{t-\tau}) + \epsilon_j^t, \tag{16}$$

$$X_j^t = \sin(X_j^{t-\tau}) + \sum_{X_i^{t-\tau} \in Pa(X_j^t) \setminus X_j^{t-\tau}} [\frac{1}{5} \times \sin(X_i^{t-\tau})] + \epsilon_j^t,$$

$$X^0 \sim \mathcal{N}(0, 1), \epsilon^t \sim \mathcal{N}(0, 0.4), \tag{17}$$

Results of the experiments on additional structural equations are presented in Table 5.

As the results shown in Table 5, our CIT-TBP achieves excellent performance in both linear and non-linear cases. Combined with the previous results in Table 1, we can deduce that our method is applicable to a wide range of different linear and nonlinear scenarios.

## D.2 Experiments on Different Numbers of Nodes and Edges

To verify the scalability of our algorithm, CIT-TBP, we fixed the number of nodes to 10 and varied the probability of generating edges between every two nodes, set p=0.1, 0.3, and 0.5. The results

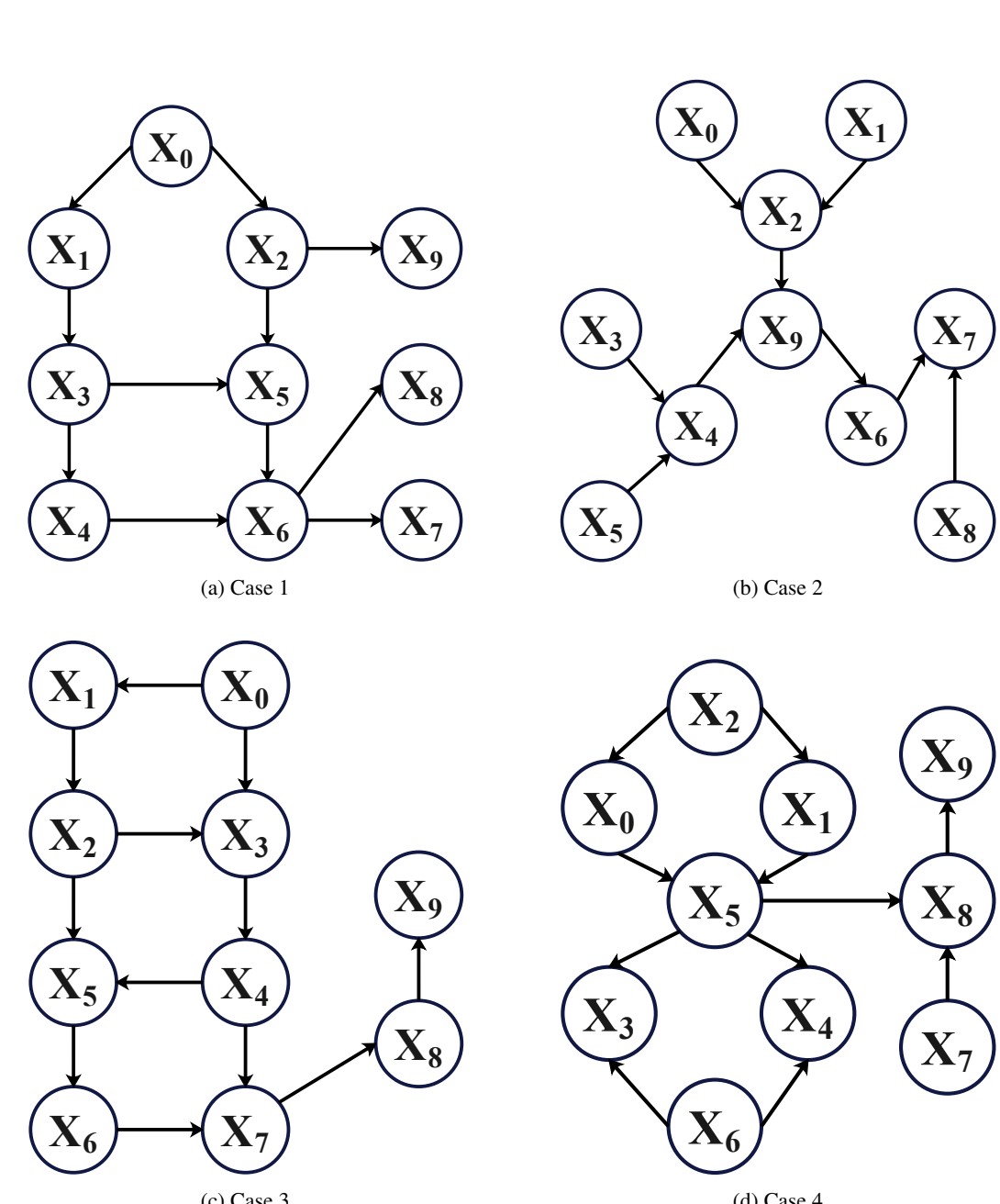

(a) Case 1

(b) Case 2

(c) Case 3

(d) Case 4

Figure 3: Causal graphs of different cases

Table 5: Results on different causal structures. (Additional equations.)

(a) Linear results.

| Algorithm | Case1 | | | Case2 | | | Case3 | | | Case4 | | |
|---|---|---|---|---|---|---|---|---|---|---|---|---|
| | SHD (↓) | F1 (↑) | FDR (↓) | SHD (↓) | F1 (↑) | FDR (↓) | SHD (↓) | F1 (↑) | FDR (↓) | SHD (↓) | F1 (↑) | FDR (↓) |
| GSP | 11.0 | 0.652 | 0.423 | 9.0 | 0.667 | 0.350 | 11.0 | 0.625 | 0.423 | 25.0 | 0.444 | 0.636 |
| CUTS+ | 5.0 | 0.851 | 0.259 | **8.0** | 0.762 | 0.304 | 12.0 | 0.772 | 0.371 | 20.0 | 0.458 | 0.593 |
| GC | 29.0 | 0.541 | 0.630 | 16.0 | 0.667 | 0.469 | 27.0 | 0.603 | 0.569 | 34.0 | 0.358 | 0.739 |
| GOLEM | 8.0 | 0.833 | 0.286 | 11.0 | 0.681 | 0.429 | 4.0 | 0.880 | 0.214 | 20.0 | 0.449 | 0.607 |
| RHINO | 8.0 | 0.833 | 0.286 | 12.0 | 0.708 | 0.414 | 13.0 | 0.772 | 0.371 | 26.0 | 0.436 | 0.647 |
| DAGMA | 8.0 | 0.800 | 0.200 | **8.0** | **0.769** | **0.250** | 8.0 | 0.810 | 0.150 | 20.0 | 0.488 | 0.500 |
| SCORE | **2.0** | 0.900 | 0.100 | 9.0 | **0.769** | **0.250** | 7.1 | 0.810 | 0.150 | 20.0 | 0.488 | 0.500 |
| PO-LINGAM | 10.0 | 0.737 | 0.222 | 11.0 | 0.667 | 0.350 | 6.0 | 0.857 | 0.100 | **15.0** | **0.585** | **0.400** |
| LPCMCI+ | 3.0 | 0.930 | 0.130 | 9.0 | 0.762 | 0.304 | **0.0** | **1.000** | **0.000** | 17.0 | 0.558 | 0.455 |
| Ours | **2.0** | **0.952** | **0.091** | **8.0** | **0.769** | **0.250** | 4.0 | 0.909 | 0.100 | 17.0 | 0.558 | 0.455 |

(b) Non-linear results.

| Algorithm | Case1 | | | Case2 | | | Case3 | | | Case4 | | |
|---|---|---|---|---|---|---|---|---|---|---|---|---|
| | SHD (↓) | F1 (↑) | FDR (↓) | SHD (↓) | F1 (↑) | FDR (↓) | SHD (↓) | F1 (↑) | FDR (↓) | SHD (↓) | F1 (↑) | FDR (↓) |
| GSP | 18.0 | 0.593 | 0.529 | 14.0 | 0.694 | 0.433 | 21.0 | 0.542 | 0.568 | 29.0 | 0.436 | 0.647 |
| CUTS+ | 14.0 | 0.702 | 0.460 | 11.0 | 0.708 | 0.414 | 14.0 | 0.746 | 0.405 | 21.0 | 0.453 | 0.625 |
| GC | 21.0 | 0.588 | 0.583 | 15.0 | 0.667 | 0.448 | 30.0 | 0.564 | 0.607 | 28.0 | 0.441 | 0.658 |
| GOLEM | 8.0 | 0.833 | 0.286 | 12.0 | 0.681 | 0.429 | 8.0 | 0.800 | 0.286 | 18.0 | **0.571** | 0.500 |
| RHINO | 8.0 | 0.833 | 0.286 | 8.0 | **0.773** | 0.320 | 5.0 | 0.898 | 0.185 | 17.0 | 0.511 | 0.538 |
| DAGMA | 7.0 | 0.762 | 0.273 | **7.0** | 0.757 | 0.222 | 10.0 | 0.667 | 0.385 | 17.0 | 0.537 | 0.450 |
| SCORE | 11.4 | 0.708 | 0.393 | 13.2 | 0.636 | 0.440 | 13.7 | 0.654 | 0.433 | 23.0 | 0.500 | 0.581 |
| PO-LINGAM | 13.0 | 0.682 | 0.375 | 10.0 | 0.700 | 0.333 | 14.0 | 0.651 | 0.333 | 18.0 | 0.524 | 0.476 |
| LPCMCI+ | **6.0** | **0.870** | 0.231 | 14.0 | 0.681 | 0.429 | **4.0** | **0.917** | 0.154 | 22.0 | 0.478 | 0.560 |
| Ours | 7.0 | 0.800 | **0.067** | 8.0 | 0.706 | **0.200** | 7.0 | 0.811 | **0.000** | **16.0** | 0.541 | **0.375** |

**Bold** indicates the best performance, underline indicates the second-best.

are shown in Table 6, respectively. From the results, we can know our method performs the best on average across the number of nodes and sparsity, but there is still some room for improvement in performance on large and dense graphs.

Table 6: Results on different graph densities.

(a) Linear causality

| Algorithm | $p = 0.1$ | | | $p = 0.3$ | | | $p = 0.5$ | | |
|---|---|---|---|---|---|---|---|---|---|
| | SHD ($\downarrow$) | F1 ($\uparrow$) | FDR ($\downarrow$) | SHD ($\downarrow$) | F1 ($\uparrow$) | FDR ($\downarrow$) | SHD ($\downarrow$) | F1 ($\uparrow$) | FDR ($\downarrow$) |
| GSP | 9.0 | 0.710 | 0.313 | 22.0 | **0.560** | 0.440 | 28.0 | 0.484 | 0.464 |
| CUTS+ | 12.0 | 0.571 | 0.333 | 26.0 | 0.481 | 0.480 | 24.0 | 0.630 | 0.324 |
| GC | **6.0** | 0.750 | 0.294 | 25.0 | 0.542 | 0.529 | 27.0 | **0.652** | 0.473 |
| GOLEM | 12.0 | 0.647 | 0.421 | 26.0 | 0.491 | 0.536 | 24.0 | 0.548 | 0.393 |
| RHINO | 6.0 | 0.727 | 0.333 | 22.0 | 0.489 | 0.560 | 23.0 | 0.571 | 0.529 |
| DAGMA | 8.0 | 0.733 | 0.267 | 24.0 | 0.444 | 0.500 | 26.0 | 0.444 | 0.400 |
| SCORE | 8.0 | 0.733 | 0.267 | 26.4 | 0.440 | 0.560 | 27.7 | 0.522 | 0.486 |
| PO-LINGAM | 9.0 | 0.710 | 0.313 | 19.0 | 0.558 | **0.333** | 21.0 | 0.618 | 0.190 |
| LPCMCI+ | 12.0 | 0.595 | 0.500 | 25.0 | 0.480 | 0.520 | 24.0 | 0.609 | 0.400 |
| Ours | **6.0** | **0.786** | **0.154** | **18.0** | 0.524 | 0.353 | **20.0** | 0.630 | **0.150** |

(b) Non-linear causality

| Algorithm | $p = 0.1$ | | | $p = 0.3$ | | | $p = 0.5$ | | |
|---|---|---|---|---|---|---|---|---|---|
| | SHD ($\downarrow$) | F1 ($\uparrow$) | FDR ($\downarrow$) | SHD ($\downarrow$) | F1 ($\uparrow$) | FDR ($\downarrow$) | SHD ($\downarrow$) | F1 ($\uparrow$) | FDR ($\downarrow$) |
| GSP | 12.0 | 0.667 | 0.429 | 28.0 | 0.523 | 0.575 | 31.0 | 0.453 | 0.585 |
| CUTS+ | **5.0** | 0.737 | 0.417 | **15.0** | 0.571 | 0.600 | 24.0 | 0.455 | 0.706 |
| GC | 13.0 | 0.600 | 0.520 | 33.0 | 0.450 | 0.673 | 34.0 | 0.581 | 0.590 |
| GOLEM | 12.0 | 0.647 | 0.421 | 28.0 | 0.453 | 0.571 | 23.0 | 0.581 | 0.357 |
| RHINO | **5.0** | 0.737 | 0.417 | **15.0** | 0.571 | 0.600 | 24.0 | 0.455 | 0.706 |
| DAGMA | 10.0 | 0.688 | 0.353 | 27.0 | 0.453 | 0.571 | **22.0** | **0.667** | 0.313 |
| SCORE | 11.2 | 0.667 | 0.389 | 27.0 | 0.464 | 0.581 | 29.4 | 0.493 | 0.514 |
| PO-LINGAM | 8.0 | **0.750** | 0.294 | 25.0 | 0.490 | 0.500 | 26.0 | 0.517 | 0.375 |
| LPCMCI+ | 8.0 | **0.750** | 0.294 | 25.0 | 0.480 | 0.520 | 25.0 | 0.606 | 0.375 |
| Ours | 8.0 | 0.714 | **0.231** | **15.0** | **0.579** | **0.154** | **22.0** | 0.549 | **0.176** |

**Bold** indicates the best performance, underline indicates the second-best.

Then we set the probability of generating edges to $p=0.1$, and gradually increased the number of nodes $N$ in the causal graphs. Specifically, we tested the causal discovery performance on graphs with 5, 10, 20, 30, and 50 nodes. The results are shown in Table 7 and Table 9.

In addition, we calculated the average ranking of the performance of these metrics in terms of the number of nodes, and subsequently calculated the overall average ranking of several metrics under each method. The results are shown in Table 8 and Table 10.

### D.3 EXPERIMENTS ON LATENT FACTOR EXCLUSION

We consider three cases in the experiments (Jin et al., 2024; Kaltenpoth & Vreeken, 2023): Case A) latent root nodes, Case B) latent mediators between measured variables, and Case C) complex latent structures in the general case. In the main text, we showed the causal graphs and results in Case A. Here we showed the causal graphs and experiment results of Case B and Case C.

The causal graphs for Case B are shown in Figure (4a), and Case C are shown in Figure (4b). We conducted our experiments on both linear and non-linear structural equations and evaluated the performance through SHD and ES. Experimental results in Table 11 and Table 12 show that our

Table 7: Performance comparison of different methods at varying node scales (Linear)

(a) N = 5

| Method | SHD (↓) | F1 (↑) | FDR (↓) |
|---|---|---|---|
| CUTS+ | **0.0** | **1.000** | **0.000** |
| RHINO | 4.0 | 0.778 | **0.000** |
| DAGMA | 1.0 | 0.933 | **0.000** |
| SCORE | 2.0 | 0.750 | 0.250 |
| PO-LINGAM | 2.0 | 0.875 | 0.125 |
| Ours | 3.0 | 0.769 | **0.000** |

(b) N = 10

| Method | SHD (↓) | F1 (↑) | FDR (↓) |
|---|---|---|---|
| CUTS+ | **5.0** | 0.762 | 0.333 |
| RHINO | 12.0 | 0.533 | 0.333 |
| DAGMA | 8.0 | 0.733 | 0.267 |
| SCORE | 8.0 | 0.733 | 0.267 |
| PO-LINGAM | 9.0 | 0.710 | 0.313 |
| Ours | 6.0 | **0.786** | **0.154** |

(c) N = 20

| Method | SHD (↓) | F1 (↑) | FDR (↓) |
|---|---|---|---|
| CUTS+ | **18.0** | **0.667** | 0.500 |
| RHINO | 45.0 | 0.505 | 0.333 |
| DAGMA | 44.0 | 0.500 | 0.551 |
| SCORE | 37.1 | 0.537 | 0.488 |
| PO-LINGAM | 33.0 | 0.560 | 0.417 |
| Ours | 24.0 | **0.667** | **0.273** |

(d) N = 30

| Method | SHD (↓) | F1 (↑) | FDR (↓) |
|---|---|---|---|
| CUTS+ | 185.0 | 0.297 | 0.427 |
| RHINO | 199.0 | 0.295 | **0.366** |
| DAGMA | 128.0 | 0.313 | 0.730 |
| SCORE | 116.6 | 0.346 | 0.686 |
| PO-LINGAM | **68.0** | **0.521** | 0.373 |
| Ours | 83.0 | 0.440 | 0.507 |

(e) N = 50

| Method | SHD (↓) | F1 (↑) | FDR (↓) |
|---|---|---|---|
| CUTS+ | 427.0 | 0.237 | 0.570 |
| RHINO | 391.0 | 0.239 | 0.618 |
| DAGMA | 353.0 | 0.255 | 0.793 |
| SCORE | 295.2 | 0.295 | 0.739 |
| PO-LINGAM | 180.0 | **0.384** | 0.472 |
| Ours | **173.0** | 0.370 | **0.433** |

**Bold** indicates the best performance, underline indicates the second-best.

Table 8: Performance rankings of different methods. (Linear)

| Method | SHD Rank | F1 Rank | FDR Rank | Avg. Rank |
|---|---|---|---|---|
| Ours | **1** | **1** | **1** | **1** |
| PO-LINGAM | 2 | 2 | 3 | 2 |
| CUTS+ | 2 | 3 | 4 | 3 |
| SCORE | 4 | 4 | 5 | 4 |
| DAGMA | 5 | 4 | 6 | 5 |
| RHINO | 6 | 6 | 2 | 6 |

algorithm performs best in different scenarios, which also further illustrates the effectiveness of our algorithm in eliminating the influence of latent factors.

# E   THE USE OF LARGE LANGUAGE MODELS (LLMS)

The authors used a large language model (LLM) solely for writing assistance, including text polishing and formatting. The LLM did not contribute to the research methodology, analysis, or scientific conclusions.

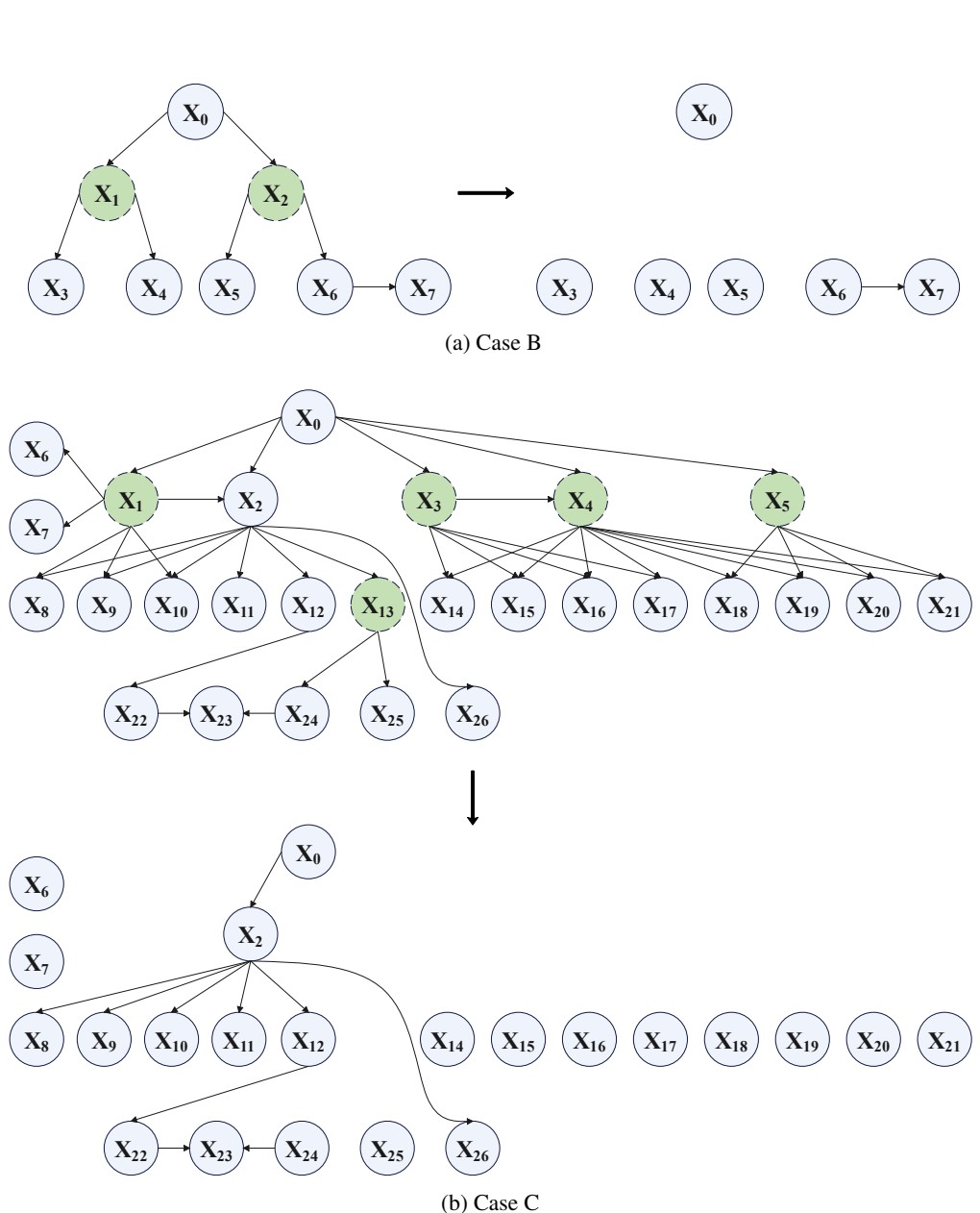

(a) Case B

(b) Case C

Figure 4: Green dashed nodes: latent factors. Blue solid nodes: observed variables. (a) Causal graphs for Case B. The left section represents the true underlying causal mechanisms, including unmeasured confounding. The right section depicts the observable variables and their relationships as available in the data. (b) Causal graphs for Case C. The upper section represents the true underlying causal mechanisms, including unmeasured confounding. The lower section depicts the observable variables and their relationships as available in the data.

Table 9: Performance comparison of different methods at varying node scales (Non-linear)

(a) N = 5

| Method | SHD (↓) | F1 (↑) | FDR (↓) |
|---|---|---|---|
| CUTS+ | 2.0 | 0.857 | 0.143 |
| RHINO | 6.0 | 0.600 | 0.143 |
| DAGMA | 6.0 | 0.588 | 0.444 |
| SCORE | 4.3 | 0.667 | 0.400 |
| PO-LINGAM | **1.0** | **0.941** | 0.111 |
| Ours | 2.0 | 0.857 | **0.000** |

(b) N = 10

| Method | SHD (↓) | F1 (↑) | FDR (↓) |
|---|---|---|---|
| CUTS+ | 13.0 | 0.514 | 0.250 |
| RHINO | 17.0 | 0.512 | **0.083** |
| DAGMA | 12.0 | 0.625 | 0.412 |
| SCORE | 13.0 | 0.629 | 0.450 |
| PO-LINGAM | **8.0** | **0.750** | 0.294 |
| Ours | **8.0** | 0.714 | 0.231 |

(c) N = 20

| Method | SHD (↓) | F1 (↑) | FDR (↓) |
|---|---|---|---|
| CUTS+ | 107.0 | 0.260 | 0.306 |
| RHINO | 51.0 | 0.418 | 0.472 |
| DAGMA | 35.0 | 0.539 | 0.462 |
| SCORE | 58.0 | 0.444 | 0.652 |
| PO-LINGAM | 39.0 | 0.519 | 0.500 |
| Ours | **25.0** | **0.606** | 0.259 |

(d) N = 30

| Method | SHD (↓) | F1 (↑) | FDR (↓) |
|---|---|---|---|
| CUTS+ | 260.0 | 0.247 | **0.341** |
| RHINO | 197.0 | 0.288 | 0.451 |
| DAGMA | 73.0 | **0.490** | 0.417 |
| SCORE | 140.7 | 0.321 | 0.741 |
| PO-LINGAM | 86.0 | 0.449 | 0.521 |
| Ours | **70.0** | 0.471 | 0.396 |

(e) N = 50

| Method | SHD (↓) | F1 (↑) | FDR (↓) |
|---|---|---|---|
| CUTS+ | 175.0 | 0.367 | 0.720 |
| RHINO | 271.0 | 0.303 | 0.661 |
| DAGMA | **166.0** | **0.384** | **0.388** |
| SCORE | 386.5 | 0.232 | 0.821 |
| PO-LINGAM | 198.0 | **0.384** | 0.547 |
| Ours | 224.0 | 0.332 | 0.644 |

**Bold** indicates the best performance, underline indicates the second-best.

Table 10: Performance rankings of different methods. (Non-linear)

| Method | SHD Rank | F1 Rank | FDR Rank | Avg Rank |
|---|---|---|---|---|
| Ours | **1** | **2** | **1** | **1** |
| PO-LINGAM | 2 | 1 | 4 | 2 |
| DAGMA | 3 | 3 | 4 | 3 |
| CUTS+ | 4 | 5 | 2 | 4 |
| RHINO | 6 | 6 | 3 | 5 |
| SCORE | 5 | 4 | 6 | 6 |

Table 11: The results on latent factor exclusion. (Case B)

| Methods | Linear SHD (↓) | Linear Error-Score (↓) | Non-linear SHD (↓) | Non-linear Error-Score (↓) |
|---|---|---|---|---|
| FCI | **1.0** | **0.0** | 1.0 | **0.0** |
| LPCMCI+ | 5.0 | 4.0 | 8.0 | 7.0 |
| PO-LINGAM | 4.0 | 4.0 | 4.0 | 3.0 |
| Ours | **1.0** | **0.0** | **0.0** | **0.0** |

**Bold** indicates the best performance.

Table 12: The results on latent factor exclusion. (Case C)

| Methods | Linear | | Non-linear | |
| --- | --- | --- | --- | --- |
| | SHD ($\downarrow$) | Error-Score ($\downarrow$) | SHD ($\downarrow$) | Error-Score ($\downarrow$) |
| FCI | 23.0 | 16.0 | 77.0 | 72.0 |
| LPCMCI+ | 34.0 | 28.0 | 71.0 | 65.0 |
| PO-LINGAM | 28.0 | 18.0 | 34.0 | **25.0** |
| Ours | **13.0** | **4.0** | **32.0** | **25.0** |

**Bold** indicates the best performance.

