# OpenReview forum: "Mitigating Confounding Effects in Causal Discovery via Time-lagged Backdoor Pathways"
_ICLR.cc/2026/Conference — ICLR 2026 Conference Withdrawn Submission_

### Official Review · Reviewer_6g97 · 2025-10-30

**Soundness:** 2
**Presentation:** 2
**Contribution:** 2
**Rating:** 2
**Confidence:** 2

**Summary:**

This paper proposes a causal-discovery method for time-series data that exploits time-lagged back-door paths to mitigate confounding effects. The method combines the conditional independence tests and  conditional transfer entropy to get the summary causal graph. The approach is evaluated on some synthetic datasets and the CausalTime benchmark to show its effectiveness.

**Strengths:**

- Strong empirical results on the CausalTime dataset.

**Weaknesses:**

- Some assumptions and theorems are described imprecisely.
- The justification for the reasonableness of the assumptions is insufficient.

**Questions:**

1. **Key challenge:** Section 2 reads like a preliminary rather than a problem statement. Please clarify what confounding means in this setting and how the reader can verify that it has been mitigated.
2. **Theorem 1:** Please define the conditioning set $C_{X_i}$ rigorously; the prose in lines 237–239 is too loose. I'm not able to catch the message in this theorem.
3. **Proof of Theorem 1**, step (a): This inference appears to contradict the Markov assumption; please reconcile.
4. **Time lag:** Experiments use a single constant $\tau$ for all variables. Can the method recover variable-specific lags? Is SHD computed on the summary graph or on the full time-series graph?
5. **Ablation:** Please report performance when using only CIT, only CTE, and their combination; explain why the combination helps.

Optional data: The telecom dataset from ICASSP 2022-SPGC root-cause analysis may serve as an additional real-world testbed.

---

### Official Review · Reviewer_Uptm · 2025-10-30

**Soundness:** 1
**Presentation:** 2
**Contribution:** 1
**Rating:** 2
**Confidence:** 5

**Summary:**

The authors propose a time-series causal discovery method CIT-TBP, focusing on mitigating confounding effects. CIT-TBP is a two-stage algorithm. First, the algorithm performs a conditional independence test on variables at time $t$ by conditioning on all other variables at time $t-\tau$ to handle observed interactions. Second, it uses conditional transfer entropy to prune remaining links, claiming this excludes spurious correlations caused by latent factors. The authors state the method is transparent and non-optimization-based, and demonstrate its performance on synthetic data.

**Strengths:**

1. Time-series causal discovery is an important problem.

**Weaknesses:**

1. The paper is confusing about its core assumptions. The existence of latent variables is not formally acknowledged until Section 3.2.2, where information entropy is introduced as a tool for latent factor exclusion. Besides, the paper never formally defines the scope of the latent variable problem it claims to solve. Can latent variables be mediators? Can they have time-lagged effects? This lack of definition makes it impossible to evaluate the theoretical claims.
2. The main text is based on a highly restrictive assumption that the system is a first-order Markov process and all causal lags are uniform and singular. It significantly limits the practical applicability of the proposed approach.
3. The two main theorems are fundamentally based on the restrictive Assumption 2. For a heterogeneous lag scenario, the conditioning strategy of Theorem 1 is no longer sufficient to block all non-causal paths. Similarly, the paper provides no discussion or proof that Theorem 2's logic holds in a heterogeneous-lag setting.
4. What is the definition of $C^{t_\tau}_{X_i}$ in Theorem 1?
5. The novelty and contribution are limited. Theorem 1 is directly based on the restrictive assumption. The use of transfer entropy is also a standard, well-known method.
5. The paper repeatedly refers to HSIC as if it were a conditional independence test. However, HSIC only measures marginal dependence. I guess what the author actually uses is the KCI test? This is a significant terminological error. Please note the expression.
6. The experiments are critically lacking heterogeneous time lags (e.g., a mix of $\tau=1$ and $\tau=2$).
7. Some sentences are misleading. For example, “constraint-based methods, such as …, typically rely on the standard backdoor criterion to uncover causal structures.”
8. Time complexity analysis is wrong.

**Questions:**

See Weakness.

---

### Official Review · Reviewer_XDQc · 2025-11-01

**Soundness:** 2
**Presentation:** 2
**Contribution:** 2
**Rating:** 4
**Confidence:** 3

**Summary:**

The paper introduces a novel approach called Conditional Independence Test on Time-Lagged Backdoor Pathways (CIT-TBP) for uncovering causal relationships. This method leverages time-lagged backdoor pathways and integrates causal information flow to effectively address the challenges posed by latent variables and intricate interactions. The authors support their algorithm with theoretical analysis and demonstrate its effectiveness through comprehensive experiments conducted on both simulated and real-world data.

**Strengths:**

1. The paper introduces a method of CIT-TBP that leverages time-lagged backdoor pathways to infer the causal relationships among variables.
2. The authors provide theoretical analysis and experimental results to demonstrate the effectiveness of the proposed method.

**Weaknesses:**

1. Conditional transfer entropy is only applicable to discrete data, and for continuous data, discretization is required, which may lead to information loss.

2. Conditional transfer entropy has limited capability to capture the direction of information flow in nonlinear scenarios, as demonstrated in the experiments.

3. The proposed method determines the presence of latent variables and the direction of relationships based on whether the conditional transfer entropy is not significantly greater than zero. However, the theoretical boundaries of this approach are not provided.

4. In the real-world data experiments, the method was not compared with approaches specifically designed for handling latent variables, such as LPCMCI.

5. Theorem 2 assumes $X_i^{t} \nrightarrow X_j^{t} | C^{t-\tau}_{X_i} $, but if $ \Delta CTE $ is not significantly greater than zero, latent variables may exist. However, it does not clearly confirm the "strong confounding effect of latent variables"—when the confounding effect between $X_i$ and $X_j$ is weak, can $\Delta CTE$ still effectively identify it? Additionally, if there are multiple latent variables interacting (e.g., two latent variables respectively influencing $X_i$ and $X_j $), does the computational logic of $\Delta CTE$ fail? The paper does not analyze these boundary scenarios theoretically, which limits the scope of application of the theorem.

**Questions:**

1. Why were non-temporal methods chosen for comparison in the experiments instead of causal discovery algorithms specifically designed for latent variable analysis in time-series data?

2. The paper assumes a uniform time lag of 1 for causal relationships. However, in real-world scenarios, the actual time lag may vary. How can the appropriate lag length be determined in practice?

3. In cases where latent variable confounding exists but no conditions satisfying the backdoor criterion can be identified to eliminate the confounding effect, how should this issue be addressed?

---

### Official Review · Reviewer_VV6k · 2025-11-01

**Soundness:** 3
**Presentation:** 2
**Contribution:** 2
**Rating:** 2
**Confidence:** 3

**Summary:**

This paper proposes a new method of causal discovery in time-lagged causation setting. This paper leverages the conditional independence relationship of nodes in previous time lag and current time point (which the authors refer to as “time lagged back door pathways”) to eliminate spurious association from causal relations discovered. The authors also propose to use conditional transfer entropy to further eliminate the possibility falsely discovered edges, where the authors argue that entropies can help detect if there is latent variables. The authors combine the two features, time lagged back door and conditional transfer entropy into a new algorithm, CIT-TBP. The authors test CIT-TBP on multiple simulated and real datasets.

**Strengths:**

Originality: the authors discovered some backdoor pathways in time-lagged causal discovery which can be used to validate if causal relations exist. This seems to be original.

Quality: in Table 1, the authors compare their proposed algorithm with 9 other baseline methods, which is very comprehensive.

Clarify: the illustration of all figures are clear.

Significance: the SHD gain from experiment results seem to be significant.

**Weaknesses:**

1.	My first concern is the assumption of Assumption 2. I would think that Markov assumption is usually about how child node is independent of grandparent nodes given grandparent. However, the Markov property in your paper is much stronger, such that it also imposes constraints on time lag. If my understanding is correct, this assumption prohibits multiple time lag values coexisting between two nodes. That is, if there is only one edge where X1 causes X2 with time lag of 3, then it is allowed by this assumption. However, if X1 causes X2 through two mechanisms, one with 1 time lag and the other with 3 time lags, then it will not be allowed. With such assumption, the fundamental finding of this paper in 3.1 seems rather trivial, as authors artificially eliminate complicated paths from the past nodes. Then if CIT-TBP is built on this very assumption, then it already has the prior knowledge that there can only be one possible time lag allowed, so it will only examine limited set of conditional independence relations compared to other constraint based method (say, PCMCI), and unsurprising it performs better because checking fewer relations reduce the possibility of type 1 and 2 errors. Therefore, it seems that the contribution of  Time-Lagged Backdoor Pathway is built upon this assumption. This assumption limits the scope of the application significantly and makes the originality rather trivial

2.	My second concern is about utilizing entropies to detect latent variables. In Theorem 2, authors imply that if improvement of entropies is “significantly greater than zero”, then there is latent factor. However, it is unclear whether the authors can define a uniform condition of “significantly greater”, otherwise this contribution is just an intuition rather than applicable tool. Furthermore, what if in true underlying causal graph X1 can only explains a small portion of variation of X2, such that all unexplained part is purely caused by some extremely distributed white noises. In this case, the improvement in entropy may still be small but at the same time there is no latent variable.

3.	My third concern is simulation setting. The choice of number 3/5,1/5 and choice of functional equation seems to be arbitrary. Also, the author doesn’t clearly explain what is “Error-Score”. Also, the author shows comprehensive comparison in Table 1 but not in other tables.

4.	My fourth concern is with the writing. The paper heavily relies on Appendices, but a good paper should be self explanatory within the page limit of this conference. Especially in Section 3.2.2, I find no clear intuition or definition of conditional transfer entropy. Also, Section 3.2.3 seems to be abrupt and may be merged with Section 3.2.2, as it is just explanation of previous theorems.

**Questions:**

1.	Following my first concern, can you explain why you define Markov assumption in this way such that it limits the number of possible time lags. At least in PCMCI related papers, such assumptions are not there. I think your Assumption 2 artificially makes the confounder elimination process easier, such that your algorithm can exploit the fact that it only needs to check a fewer set of conditional independence relationships (say, only the nodes of previous time lags -\tau, rather than all other nodes of further previous time lags of -2*\tau). Do you think you can still show the advantages of your proposed methods without such condition of not allowing different time lag values? Also, do you think it is fair to compare this algorithm with other constraint methods given those other methods need to test further combinations of conditional independence relationships?

2.	Following my second concern, what is “significantly greater than zero” in your Theorem 2? Also, suppose in true causal graph, X2 only has one true parent X1, but the relation of X1 and X2 are weak (X1 can only explain a partial variation of X2 because X2 has an extremely distributed white noise), then will the entropy improvement be close to zero such that your algorithm thinks that there is latent variable?

3.	In your simulation result Table 2 and 3, can you also show the performance of other baselines in Table 1 (say, the performance of GOLEM)? Also, in Table 1(b), Case1, LPCMCI+ seems to perform much better than your method (SHD = 1 vs SHD = 10), how to explain that? What is “Error-Score”?

---

### Note · Authors · 2025-12-09

I have read and agree with the venue's withdrawal policy on behalf of myself and my co-authors.